# A BENCHMARK STUDY ON CALIBRATION

**Linwei Tao**
University of Sydney
linwei.tao@sydney.edu.au

**Younan Zhu, Haolan Guo**
University of Sydney
{yzhu0986, hguo4658}@uni.sydney.edu.au

**Minjing Dong**
City University of Hong Kong
minjdong@cityu.edu.hk

**Chang Xu**
University of Sydney
c.xu@sydney.edu.au

## ABSTRACT

Deep neural networks are increasingly utilized in various machine learning tasks. However, as these models grow in complexity, they often face calibration issues, despite enhanced prediction accuracy. Many studies have endeavored to improve calibration performance through the use of specific loss functions, data preprocessing and training frameworks. Yet, investigations into calibration properties have been somewhat overlooked. Our study leverages the Neural Architecture Search (NAS) search space, offering an exhaustive model architecture space for thorough calibration properties exploration. We specifically create a model calibration dataset. This dataset evaluates 90 bin-based and 12 additional calibration measurements across 117,702 unique neural networks within the widely employed NATS-Bench search space. Our analysis aims to answer several longstanding questions in the field, using our proposed dataset: (i) Can model calibration be generalized across different datasets? (ii) Can robustness be used as a calibration measurement? (iii) How reliable are calibration metrics? (iv) Does a post-hoc calibration method affect all models uniformly? (v) How does calibration interact with accuracy? (vi) What is the impact of bin size on calibration measurement? (vii) Which architectural designs are beneficial for calibration? Additionally, our study bridges an existing gap by exploring calibration within NAS. By providing this dataset, we enable further research into NAS calibration. As far as we are aware, our research represents the first large-scale investigation into calibration properties and the premier study of calibration issues within NAS. The project page can be found at https://www.taolinwei.com/calibration-study.

## 1 INTRODUCTION

Despite their widespread success across various domains, deep neural networks (DNNs) are not immune to producing miscalibrated predictions, leading to either overconfidence or underconfidence. This concern becomes particularly salient for safety-critical applications such as autonomous driving (Feng et al., 2019) and medical diagnosis (Thiagarajan et al., 2022), where reliance on accurate prediction probabilities is paramount. In these contexts, miscalibrated predictions may give rise to potentially catastrophic consequences.

A myriad of attempts (Mukhoti et al., 2020; Kumar et al., 2018; Tao et al., 2023a; Karandikar et al., 2021; Krishnan & Tickoo, 2020; Zhang et al., 2022; Hendrycks et al., 2019; Müller et al., 2019; Deng & Zhang, 2021; Kim et al., 2021; Tao et al., 2023b) has been made to mitigate the issue of miscalibration, primarily focusing on loss functions or training frameworks. However, the calibration properties of the neural network architectures themselves have received comparatively less attention. Guo et al. (2017) were among the first to investigate the relationship between neural architectures and calibration performance, but their work was restricted to the effects of varying depth and width in a ResNet(He et al., 2016). More recent studies (Minderer et al., 2021) have extended this exploration to modern neural networks, such as the non-convolutional MLP-Mixer (Tolstikhin et al., 2021) and Vision Transformers (Ranftl et al., 2021). However, the limited diversity and quantity of model architectures have constrained the depth of calibration property studies.

NAS (Liu et al., 2018; Dong & Yang, 2019b;a; Xu et al., 2019; Su et al., 2022) has brought about a revolution in the field of deep learning by automating the discovery of neural architectures that outperform traditionally hand-designed architectures like AlexNet(Krizhevsky et al., 2009) and ResNet (He et al., 2016). NAS has achieved advancements in part due to its predefined comprehensive model architecture space. NAS benchmark search spaces (Ying et al., 2019; Dong & Yang, 2020; Siems et al., 2020; Dong et al., 2021) have provided an extensive set of unique convolutional architectures for NAS research. While most prior works on search space focus on the topological architecture of models, the NATS-Bench (Dong et al., 2021) provides a more expansive search space, taking into account models of different sizes. This comprehensive search space can serve as a potent tool to bridge the gap in previous calibration studies.

In our work, to exam the calibration property and address calibration-related research questions, such as the reliability of calibration metrics, one approach is to assess the consistency of different metrics based on a substantial number of well-trained models. However, collecting such a substantial dataset is often challenging due to the associated training costs. Fortunately, NATS-Bench (Dong et al., 2021) provides access to 117.9K well-trained models with various architectural designs, enabling us to conduct a comprehensive and generalisable study. Specifically, we evaluate all 117,702 unique CNN architectures concerning topology and model size, and benchmark them on multiple calibration metrics of different types. We also include the results on 11 different Vision Transformers to generalize our findings. The specifics about architectures and metrics are discussed in section 3. This comprehensive dataset has served as a benchmark for our subsequent studies and in-depth analysis on the calibration properties. In this study, we try to answer the following questions that are longstanding in this field:

1. Can model calibration be generalized across different datasets? 4.1
2. Can robustness be used as a calibration measurement? 4.2
3. How reliable are calibration metrics? 4.3
4. Does a post-hoc calibration method affect all models uniformly? 4.4
5. How does calibration interact with accuracy? 4.5
6. What is the impact of bin size on calibration measurement? 4.6
7. Which architectural designs are beneficial for calibration? 4.7

This exploration aims to shed light on the often-overlooked aspect of calibration, thereby contributing to a more holistic understanding of deep learning model performance.

## 2 RELATED WORKS

**Calibration Metrics** Extensive research has been done on calibration metrics, which is crucial for measure the reliability of predictions. The Expected Calibration Error (ECE) is a widely used metric introduced by Naeini et al. (2015), which quantifies the absolute difference between predicted confidence and empirical accuracy. However, ECE is susceptible to estimator bias, complicating its estimation (Nixon et al., 2019; Vaicenavicius et al., 2019; Gupta & Ramdas, 2021). To address the bias-variance trade-off, adaptive binning techniques such as Equal Mass Binning calibration error (ECE$em$) have been proposed by Kumar et al. (2019) and Nixon et al. (2019). In multi-class settings, class-wise calibration errors, including Classwise Equal-Width Binning calibration error (cwCE) and Classwise Equal-Mass Binning calibration error (cwCE$em$), offer potential solutions (Kumar et al., 2019). Alternatives to ECE encompass likelihood-based measures, the Brier score (Brier, 1950), and Bayesian methods (Gelman & Shalizi, 2013). Zhang et al. (2020) proposed the Top-Label calibration error using Kernel Density Estimation (KDECE) to avoid binning schemes. Gupta et al. (2020) introduced the Kolmogorov-Smirnov calibration error (KSCE), which uses the Kolmogorov-Smirnov test to compare empirical cumulative distribution functions. Additionally, Kumar et al. (2018) suggested the Maximum Mean calibration error (MMCE), a differentiable estimator that compares the Top-Label confidence with the conditional accuracy.

**Empirical Study on Calibration** There have been limited studies on evaluating the calibration of uncertainty estimates in deep learning models. Guo et al. (2017) discovered that many models with deeper or wider architectures are poorly calibrated and suggested a simple post-processing method called temperature scaling that can substantially improve calibration. In a recent study, Minderer

et al. (2021) revisited the calibration of modern neural networks, such as MLP-mixer (Tolstikhin et al., 2021) and ViT (Dosovitskiy et al., 2020b), and found that these new, larger models outperform CNNs. They also examined the impact of training frameworks like SimCLR (Chen et al., 2020) and CLIP (Radford et al., 2021). However, these studies are based on a limited set of model architectures, and more research is needed to generalize their findings.

**NAS** The search space in NAS serves as a critical component for exploring calibration properties. Various search spaces, such as fully connected, convolutional, recurrent, and attention-based architectures, have been proposed in existing literature. NAS-Bench-101 (Ying et al., 2019), the inaugural public architecture dataset for NAS research, comprises 423k unique convolutional architectures. NAS-Bench-201 (Dong & Yang, 2020) expands on NAS-Bench-101 by offering a different search space. NAS-Bench-301 (Siems et al., 2020) addresses the issues found in tabular NAS benchmarks. Most prior studies on search spaces concentrate on the topological architecture of models. Recently, NATS-Bench (Dong et al., 2021) was introduced as an extension to NAS-Bench-201 (Dong & Yang, 2020), providing an expanded search space for models of varying sizes. This expansion marks a significant advancement in the field, creating new possibilities for the exploration and improvement of deep learning models.

## 3 DATASET GENERATION

In this section, we detail the metrics and model architectures from NATS-Bench (Dong et al., 2021) that are involved in our study. To ensure the generality of our findings, we also include 11 vision transformers of different architecture design. Each unique architecture is pretrained for 200 epochs on three benchmark datasets: CIFAR-10 (Krizhevsky et al., 2009), CIFAR-100 (Krizhevsky et al., 2009), and ImageNet16-120 (Chrabaszcz et al., 2017). Note that ImageNet16-120 is a down-sampled variant of ImageNet, widely used in NAS literature (Ying et al., 2019; Dong & Yang, 2020; Dong et al., 2021; Patel et al., 2020). We choose this dataset because it can reduce computational costs while maintaining comparable results as ImageNet (Chrabaszcz et al., 2017). For simplicity, we refer to ImageNet16-120 as ImageNet in the following discussion. To evaluate post temperature scaling, we create a validation set by splitting the original test set into 20%/80% for validation/test.

We use extensive bin-based metrics and a range of other calibration measures in this study, which provides a thorough assessment of model calibration performance across different scenarios. We evaluate all calibration metrics discussed in 2 and more details about metrics can be found in Appendix J. Since prediction performance on Out-of-Distribution (OoD) datasets is also a strong indicator of calibration (Mukhoti et al., 2020), we include the Area Under the Curve (AUC) for the TSS models, evaluating on two OoD datasets, CIFAR-10-C (corrupted with Gaussian noise) (Hendrycks & Dietterich, 2019) and SVHN (Street View House Numbers) (Netzer et al., 2011). We evaluate these metrics across a wide range of bin sizes, including 5, 10, 15, 20, 25, 50, 100, 200, and 500 bins. These metrics are assessed both before and after temperature scaling. Overall, this results in the assessment of 102 different measurements, providing a comprehensive evaluation of model calibration.

### 3.1 ARCHITECTURES EVALUATED

We evaluate calibration properties on 117,702 unique architectures in NATS-Bench (Dong et al., 2021), which is a cell-based neural architecture search space consisting of two search spaces: Topology Search Space (TSS) and Size Search Space (SSS). Referring to Figure 1, each cell is represented by a densely-connected directed acyclic graph with four nodes and six edges. Within this structure, nodes denote feature maps, while edges correspond to selected operations from a predefined operation set $O$. The operation set $O$ encompasses the following operations: $O = \{1 \times 1 \text{ convolution}, 3 \times 3 \text{ convolution}, 3 \times 3 \text{ average pooling}, \text{skip}, \text{zero}\}$. The search space contains 15,625 architectures, but only 6,466 are unique due to isomorphic cells resulting from the skip and zero operations.

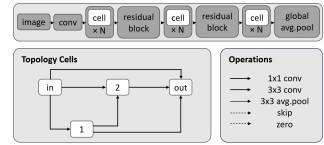

Figure 1: the macro skeleton of each candidate architecture at the top, cell representations at the bottom-left, and operation candidates at the bottom-right. The candidate channels for SSS are 8, 16, 24, 32, 40, 48, 56, and 64.

In SSS, each architecture has a different configuration for the number of channels in each layer. In this space, "stacks" refer to the aggregation of multiple cells. Each stack is constructed using the cell structure that has demonstrated the best performance in the TSS on the CIFAR-100 dataset. A total of 32,768 architecture candidates are evaluated on the same three

datasets for 90 epochs. For our evaluations, we assess all architectures within both search spaces and evaluate them on different test splits of their corresponding datasets. In total, we evaluate a total of $3 \times 6466 = 19398$ networks on TSS and $3 \times 32768 = 98304$ on SSS.

To ensure the generality of our findings, we also include 11 vision transformers of different architecture design including T2T-ViT-7, T2T-ViT-10, T2T-ViT-12, T2T-ViT-19, T2T-ViT-24 (Yuan et al., 2021), ViT-b-16 (Dosovitskiy et al., 2020a), Swin-T (Liu et al., 2021), Deit-T (Touvron et al., 2021a), Cait-XXS24 (Touvron et al., 2021b), PvTv2-T (Wang et al., 2022) and PoolFormer-S12 (Yu et al., 2022) . Each transformer is fine tuned 60 epochs on CIFAR-10, CIFAR-100 and ImageNet-16-120 based on the pretrained weights on ImageNet-1k.

## 4 EXPERIMENTS AND DISCUSSION

Our research endeavors to investigate the influence of several factors, including the dataset, calibration metrics, post-hoc calibration methods, accuracy and robustness on the calibration performance of neural networks. Additionally, we seek to analyze the impact of the architecture designs on calibration. With the comprehensive evaluation results in hand, we have made several empirical observations and derived insightful conclusions, which could be summarized to answer the following questions.

### 4.1 CAN MODEL CALIBRATION BE GENERALIZED ACROSS DIFFERENT DATASETS?

We're particularly interested in understanding if there are noticeable variations in model calibration when different datasets are used for training. Additionally, we aim to investigate how the complexity and diversity of the dataset might impact calibration performance.

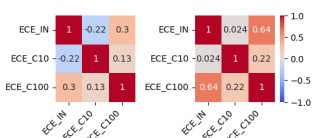

**Experimental setup:** We examine two subsets of ECE measurement from our TSS dataset: the first subset consists of architectures with good prediction accuracy in each dataset (ranked top 1000 by accuracy), while the second subset includes all architectures available in the dataset. The aim of our investigation is to probe the model calibration performance correlation between datasets. To achieve this, we resort to the Kendall ranking correlation coefficient(Kendall, 1938). This non-parametric statistic is a potent tool for measuring the degree of correspondence between two rankings, with its values ranging from -1 (indicating perfect disagreement) to +1 (indicating perfect agreement). A Kendall coefficient of zero would signify a lack of correlation. With the help of Kendall ranking correlation co-

Figure 2: Kendall ranking correlation matrix of ECE for different TSS architecture subsets. The left subplot corresponds to the top 1000 architectures based on accuracy, while the right subplot represents the entire set of models.

efficient, we build a correlation matrix for ECE on CIFAR-10 (ECE_C10), CIFAR-100 (ECE_C100), ImageNet (ECE_IN) for both models with high accuracy (left) and all models (right) as displayed in Figure 2.

**Discussion:** Upon examining the correlation across datasets, we observed a substantial variation in the ranking of calibration metrics. Notably, the calibration performance on CIFAR-10 exhibited little correlation with the calibration performance on CIFAR-100, despite their images being the same. Furthermore, the calibration performance on CIFAR-10 exhibited no correlation with the performance on ImageNet and even negatively correlated for top 1000 models. In contrast, the correlation of calibration performance between the more complex datasets CIFAR-100 and ImageNet was relatively higher. This observation suggests that the relationship between calibration performance of a certain model and the evaluation dataset may be weak or non-existent. Thus, **the calibration property of a certain architecture can not generalize well to different datasets**, researchers cannot rely on evaluation results on different datasets when selecting a calibrated architecture for downstream datasets. We draw the similar conclusion among Transformer architectures as shown in Appendix A. Our observation is also consistent among other calibration metrics, as detailed in the Appendix K.

### 4.2 CAN ROBUSTNESS BE USED AS A CALIBRATION MEASUREMENT?

Previous research (Thulasidasan et al., 2019) suggests that a well-calibrated model should perform well on OoD datasets. Several studies, including (Mukhoti et al., 2020), have utilized this notion to evaluate their calibration methods. Given that accuracy on corruption datasets is a robustness indicator and have a strong correlation with adversarial robustness, a question arises: can other robustness metrics serve as a measure of calibration?

**Experiment Setup:** To address this question, we focus on the Kendall ranking correlation matrix between ECE and various metrics for models ranked in the top 1000, 2000, and so on, based on their accuracy. Recently, a dataset based on NAS-Bench-201 (Jung et al., 2023) was proposed, which evaluates the robustness performance on all TSS models available in NATS-Bench and enables us to evaluate the relationship between robustness and calibration. Thus, the consid-

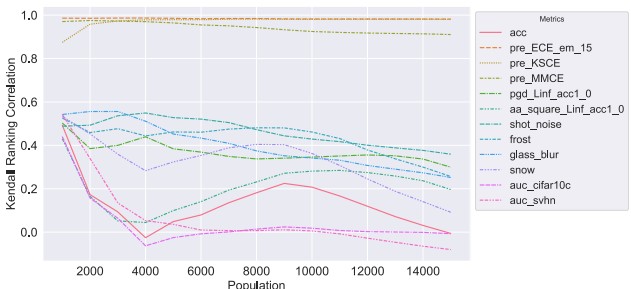

Figure 3: Kendall ranking correlation of various metrics against ECE different top-ranked model population.

ered metrics include accuracy, calibration metrics such as bin-based $ECE_{em}$, statistics-based KSCE, and kernel-based MMCE, as well as robustness metrics like adversarial robustness (PGD attack and square attack) and accuracy on corruption datasets (forest, shot noise, snow, and glass blur), and Area under the Curve (AuC) on CIFAR-10-C and SVHN. We depicted the correlation coefficients for different populations using a line plot, as shown in Figure 3. The experiment was conducted on CIFAR-10, and experiments for other datasets are provided in Appendix N.

**Discussion:** Upon analyzing the top 1000 ranked models, we found that all robustness metrics, including adversarial robustness and accuracy on corruption datasets, exhibited a strong correlation with ECE (Jung et al., 2023). Additionally, even prediction accuracy had a high correlation with ECE. However, as we included worse-performing models, such as top 4000 accuracy models, the correlation was not observed for most metrics such as accuracy and snow corruption accuracy. Nonetheless, some robustness metrics, such as PGD attack robustness accuracy, still displayed a high correlation with ECE. On the other hand, bin-based $ECE_{em}$, statistics-based KSCE, and kernel-based MMCE had a high correlation with ECE regardless of the model's performance on prediction accuracy.

The study reveals that **calibration performance can be measured not only by the robustness accuracy on the corruption dataset, but also by other robustness metrics only among models with high prediction accuracy**. However, when considering models with varying prediction performance, there seems to be no correlation between the AuC on OoD datasets and ECE, as with most other robustness metrics. This suggests that **including AuC on OoD datasets in robustness metrics may not reliably measure calibration performance for models of varying prediction performance**. It is worth more caution when using AuC as a metric for evaluating the calibration of neural networks.

### 4.3 HOW RELIABLE ARE CALIBRATION METRICS?

We aim to explore whether certain metrics are particularly effective in specific scenarios, or if their assessments generally align. A critical aspect of this investigation involves identifying if there is a substantial disparity in the results when employing different calibration metrics.

**Experimental Setup.** We provide a Kendall ranking correlation matrix between 7 calibration metrics of different types within the CIFAR-10 and ImageNet datasets on TSS to demonstrate ranking correlations between these metrics.

**Discussion:** Despite the availability of numerous calibration metrics, a theoretical gap exists between bin-based, kernel-based, and statistics-based metrics. Our evaluation reveals the correlation between metrics, as illustrated in Figure 4a, **a consistent trend in the ranking of most calibration performance regardless of metric type**. Although multiple works (Nixon et al., 2019; Kumar et al., 2019; Roelofs et al., 2022; Gupta et al., 2020) point out the demerits of ECE, the ECE shows consistent results with most other metrics. Our analysis of the correlation coefficients of NLL and Brier score showed consistency across various metrics. We extended our correlation analysis to all models in SSS and observed similar trends. It is worth noting that the classwise-based calibration error metrics, such as cwCE and $cwCE_{em}$, exhibited a lower correlation with other metrics, where $cwCE_{em}$ is even negatively correlated with other widely accepted metrics on ImageNet. This indicates that **$cwCE_{em}$ may not be a reliable metric for calibration measurement.** We postulate that this discrepancy could stem from inherent limitations associated with equal-mass binning. Specifically, equal-mass binning might yield a bin that covers a broad range of high uncertainty. This could prioritize smaller quantization errors for low-ranked logits over focusing on high-ranked logits, which could subse-

quently undermine calibration performance. We observed that this degradation caused by equal-mass binning tends to diminish when datasets have fewer classes. For instance, in the case of CIFAR-10, the higher class prior appears to mitigate the negative effects of equal-mass binning. We draw the similar conclusion among Transformer architectures as shown in Appendix A.

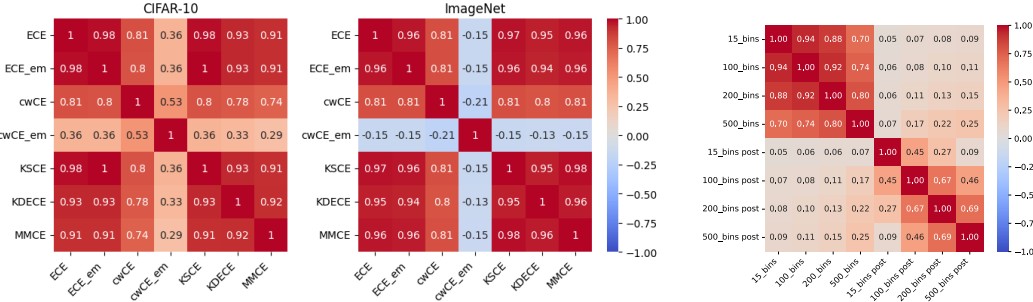

(a) Kendall ranking correlation between diverse calibration metrics. The metrics are evaluated across the entire set of TSS models. The analysis spans each of the CIFAR-10(left) and ImageNet(right).

(b) Kendall Ranking Correlation Matrix of ECE before and after temperature scaling on CIFAR-10.

Figure 4: Explore the properties of calibration metrics.

### 4.4 DOES A POST-HOC CALIBRATION METHOD AFFECT ALL MODELS UNIFORMLY?

The research conducted by Wang et al. (2021) suggests that regularized models typically generate more accurate and well-calibrated predictions, but may have lower calibratability. In other words, regularized models have lower calibration space for post-hoc calibration techniques, such as temperature scaling and histogram binning. However, this observation is based on a limited number of cases. Our dataset allows for a more comprehensive exploration of this issue.

**Experiment Setup:** To investigate this issue thoroughly, we computed the Kendall ranking correlation matrix between pre-temperature-scaling ECE (pre-ECE) and post-temperature-scaling ECE (post-ECE) on CIFAR-10 with different bin sizes for all models in TSS dataset.

**Discussion:** As shown in Figure 4b, the correlation coefficient between pre-ECE and post-ECE is nearly zero, suggesting that a well-calibrated model may not maintain its ranking of calibration performance after undergoing post-hoc calibration methods. On the other hand, a less well-calibrated model may improve its calibration performance after such methods. Our findings expand on the conclusion for regularized models and indicate that **well-calibrated models do not necessarily exhibit better calibration performance after post-hoc calibration techniques**. This observation is align with (Ashukha et al., 2020), which indicates the comparison of calibration performance between different methods without post-calibration might not provide a fair ranking. Furthermore, we note that the selection of bin size has a greater influence on post-hoc calibration measurements. Additionally, we observe similar results present on ImageNet and CIFAR-100 for both top 1000 models and all models. We draw the similar conclusion among Transformer architectures as shown in Appendix A. Additional results are included in the Appendix M.

### 4.5 HOW DOES CALIBRATION INTERACT WITH ACCURACY?

Extensive research in the robustness literature has investigated the trade-off between accuracy and robustness such as (Zhang et al., 2019). However, the interplay between accuracy and calibration has not been as thoroughly explored. While some prior studies hinted at a potential trade-off between accuracy and calibration (Mukhoti et al., 2020; Karandikar et al., 2021), these findings seem limited or not universally consistent. Therefore, further investigation into this topic using our dataset could yield valuable insights.

**Experiment Setup:** We created a scatter plot for all models and models with high accuracy on ECE versus Accuracy. The experiment was conducted on CIFAR-10. Experiments for other datasets are included in the Appendix D.

**Discussion:** Figure 5 illustrates that when considering only models with accuracy over 90% (as depicted in the left plot), a distinct pattern emerges: higher accuracy is associated with better calibration performance, whereas lower accuracy results in poorer calibration performance. However,

when analyzing all models in TSS, this pattern is not evident. Therefore, it can be concluded that **the positive correlation between accuracy and calibration exists only among architectures with good prediction performance, challenging the previously hinted trade-off.**

### 4.6 WHAT IS THE IMPACT OF BIN SIZE ON CALIBRATION MEASUREMENT?

A pivotal factor in bin-based calibration measurement is the choice of bin size (number of bins). Larger bin sizes encapsulate a wider range of confidence levels, which can potentially mask variations and result in an underestimation of miscalibration. Conversely, very small bin sizes may lead to overfitting or unstable estimates due to insufficient data in each bin. Therefore, the study on bin size choice is essential for accurate calibration measurements.

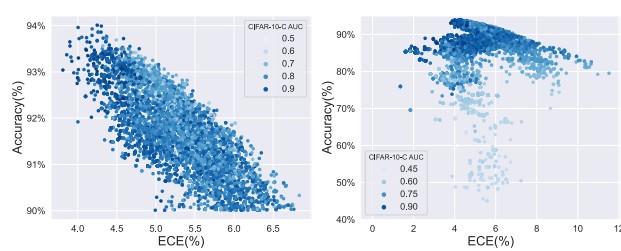

Figure 5: Scatter plots depict the ECE versus Accuracy of model with accuracy larger than 90% (left) and all TSS models (right) on CIFAR-10. The color-coded markers represent CIFAR-10-C AUC scores.

**Experimental Setup:** To address this issue, we focus on the evaluation of the bin-based calibration metrics with different bin sizes on before and after temperature scaling. We also explore the calibration performance on both 12 and 200 epochs to examine the effect of bin size on under-fitting models.

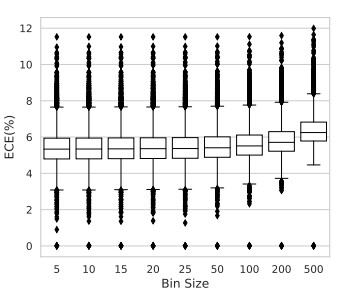
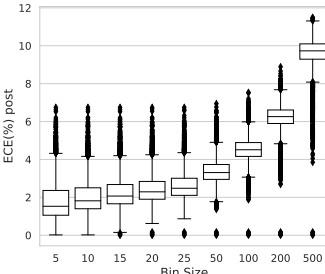
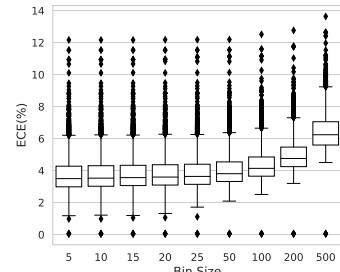

(a) Boxplot of ECE of different bin size before temperature scaling on 200 epochs.

(b) Boxplot of ECE of different bin size after temperature scaling on 200 epochs.

(c) Boxplot of ECE of different bin size before temperature scaling on 12 epochs.

Figure 6: Explore the impact of bin size on calibration before and after temperature scaling.

**Discussion:** The results from Figures 6a and 6b demonstrate that both pre- and post-ECE increase in correlation with bin size. However, the post-ECE showcases a more pronounced rise—doubling at a bin size of 100 and quintupling at a bin size of 500. This observation is further substantiated by Figure 4b, where post-ECE measurements with a bin size of 500 reveal barely any correlation when compared to those at a bin size of 100. Figures 6c and 6a display the variation of ECE across diverse bin sizes at different stages of model training. Remarkably, for models trained for only 12 epochs (shown in Figure 6c), the choice of bin size significantly influences calibration performance more than fully trained models that have trained 200 epochs.

These findings suggest that **bin size has a more substantial impact on post-ECE**, with a negligible correlation between post-ECE measurements at varying bin sizes. Conversely, pre-ECE seems more resistant to changes in bin sizes. Therefore, for a holistic comparison, **it is recommended to assess post-hoc calibration performance across a range of bin sizes.** Moreover, the influence of bin size is more conspicuous for underfitting models, which implies that choosing the correct bin size is particularly crucial for underfitting models. These trends were also observed when evaluating other bin-based calibration metrics and in the SSS, details of which are provided in the Appendix L.

### 4.7 WHICH ARCHITECTURAL DESIGNS ARE BENEFICIAL FOR CALIBRATION?

The focus now shifts towards comprehending the influence of architecture design on calibration performance. Specifically, the number of kernels and the topological architecture design are investigated. Previous research (Guo et al., 2017) has suggested that complex neural networks tend to be poorly calibrated. However, our findings only consistent with this observation in limited cases.

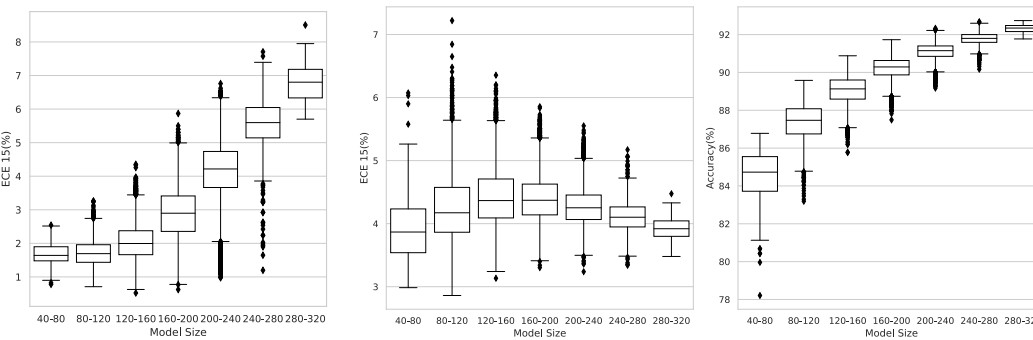

(a) Box-plots of ECE across all architectures within SSS subset on ImagenNet, segregated by different size brackets.

(b) Boxplots illustrating ECE and Accuracy across all architectural within SSS subset, segregated by different size brackets. Each plot represents the performance evaluation on CIFAR-10. The "model size" is defined as total number of kernels in each layer.

Figure 7: Explore the impact of model size on calibration performance.

**Experiment Setup:** In this study, we focus on our SSS dataset for ImageNet and CIFAR-10, segmented by different size brackets based on the total number of kernels in each layer, as shown in Figure 7. Additionally, the performance of models of different width on CIFAR-10, CIFAR-100, and ImageNet was plotted to compare the impact of model width on different datasets, as shown in Figure 9. We also choose top-ranked model in terms of our proposed metric to analyze the architecture design preference in Figure 10.

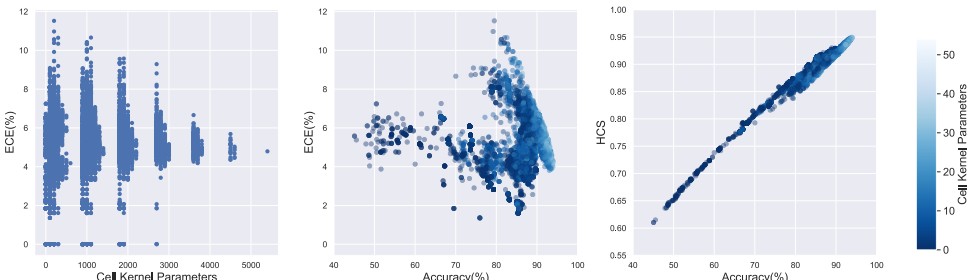

Figure 8: Calibration performance with cell kernel parameters. (Left) ECE distribution across different cell kernel parameters; (Middle) Scatter plot of all models on ECE and Accuracy; (Right): Scatter plot of all models on HCS and Accuracy. $\beta$ is set to 1.

**Discussion:** As shown in Figure 7, the results show that larger models do not necessarily have worse calibration performance, as it initially declines for the first few size brackets, but subsequently improves with increasing model size for CIFAR-10, while it worsens for ImageNet. Therefore, **calibration performance may depend on both model size and dataset complexity.** Additionally, the study found that fewer parameters do not necessarily lead to better calibration performance, as a larger kernel size does not necessarily result in worse calibration performance but instead results in a smaller calibration performance variance as depicted in the left plot of Figure 8. The conclusion of a prior study that calibration decreases with model size may be due to the survivorship bias in calibration, where architectures with worse performance are often ignored, and the literature overlooks the fact that **less parameters do not necessarily lead to better calibration performance.** Following our examination of the overall model size, we next direct our attention towards understanding the impact of the model's width, specifically the number of filters per layer, on its calibration properties. As depicted in Figure 9, on the CIFAR-10 dataset, the width of the model appears to have a modest impact on calibration. However, as the dataset complexity increases, particularly with ImageNet,

we observe a dramatic surge in ECE, rising from 3% to a substantial 8%. This suggests that **wider model can worsen calibration performance, especially when dealing with complex datasets.**

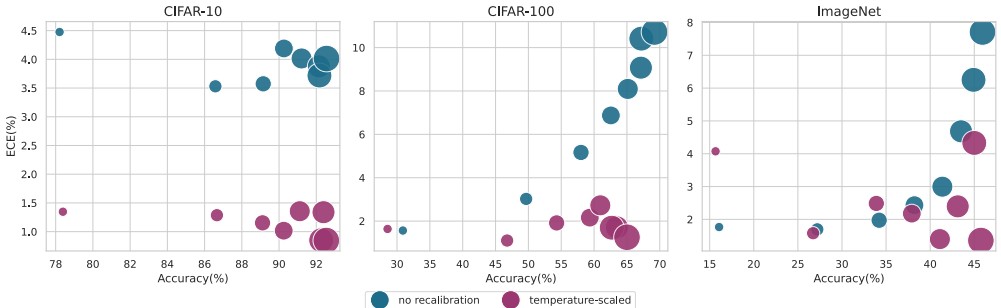

Figure 9: ECE measured on CIFAR-10, CIFAR-100, and ImageNet datasets before and after applying temperature scaling. Marker size represents model size progression from 8:8:8:8:8, 16:16:16:16:16, up to 64:64:64:64:64, where number indicate the number of kernels in a certain layer.

To analyze the best calibrated model, it is not appropriate to directly choose the model with the lowest ECE, since it may have unsatisfactory prediction accuracy performance. To address this issue, we propose a combined metric that takes into account both calibration and accuracy, called the Harmonic Calibration Score (HCS), denoted as $\mathrm{HCS}_\beta = (1 + \beta) \cdot \frac{(\mathrm{Acc} \cdot (1 - \mathrm{ECE}))}{(\beta \mathrm{Acc} + (1 - \mathrm{ECE}))}$, where $\beta$ controls the balance between accuracy and calibration with larger $\beta$ prefer more on calibration. Figure 8

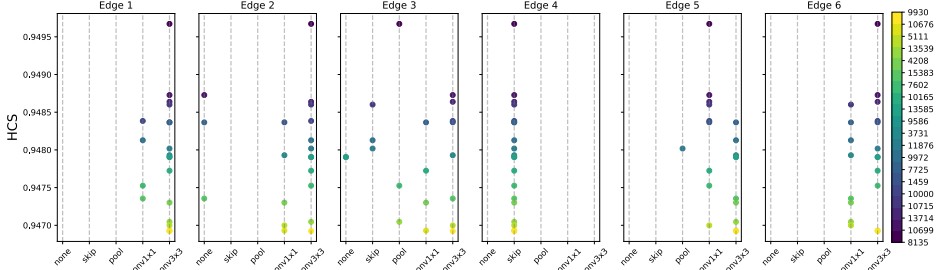

Figure 10: Top 20 HCS architectures out of the topology search space according to the ECE on CIFAR-10 dataset. $\beta$ is set to 1.

illustrates the distribution of models with respect to ECE and accuracy, where the best ECE is not always obtained at the same time as the highest accuracy. However, our proposed HCS provides a good trade-off between accuracy and calibration, and the model with the highest HCS indicates a better balance between calibration and prediction accuracy. After applying the HCS metric and replotting, a clear trend can be observed where the HCS increases as the cell kernel parameters increase. We subsequently selected the top 20 models based on their HCS scores from TSS and performed an architectural analysis of each edge, as shown in Figure 10. This analysis shows that well-calibrated models tend to prefer either $1 \times 1$ convolution or $3 \times 3$ convolution for edges 1, 5, and 6. Intriguingly, we observed a unanimous preference for the skip connection for edge 4 among these well-calibrated models. These observations offer valuable insights into the design preferences of well-calibrated models, which can guide the development of future neural network architectures.

## 5 CONCLUSIONS

While our investigation provides significant insights into calibration studies, it is essential to note its specific scope. Our findings predominantly relate to image classifications, implying that extrapolation to other domains should be approached with caution. Within our analysis, we emphasized the post-hoc Temperature Scaling technique; however, the vast domain of post-hoc calibration houses numerous other techniques that may present divergent perspectives or different effects on model calibration. Importantly, our conclusions are primarily empirical. A theoretical exploration remains outside the purview of this work. Future research could remedy these confines by widening the spectrum of evaluated tasks and architectures and diving deeper into theoretical dissections.

ACKNOWLEDGEMENTS

We thank Yang You and Yifeng Gao for valuable discussions and feedbacks. This work was supported in part by the Australian Research Council under Projects DP240101848 and FT230100549.

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
