## A  EXPERIMENTS ON TRANSFORMERS

To ensure the generality of our findings, we also include 11 vision transformers of different architecture design including T2T-ViT-7, T2T-ViT-10, T2T-ViT-12, T2T-ViT-19, T2T-ViT-24 (Yuan et al., 2021), ViT-b-16 (Dosovitskiy et al., 2020a), Swin-T (Liu et al., 2021), Deit-T (Touvron et al., 2021a), Cait-XXS24 (Touvron et al., 2021b), PvTv2-T (Wang et al., 2022) and PoolFormer-S12 (Yu et al., 2022) . Each transformer is fine tuned 60 epochs on CIFAR-10, CIFAR-100 and ImageNet-16-120 based on the pretrained weights on ImageNet-1k.

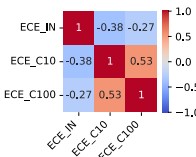

Figure 11: Kendall ranking correlation matrix of ECE for different Vision Transformer architectures.

As shown in Figure 11, transformers show similar results as CNNs, where ECE on ImageNet shows little correlation with that on CIFAR-10 and CIFAR-100. The observation indicates that calibration property of a certain architecture can not generalize well to different datasets.

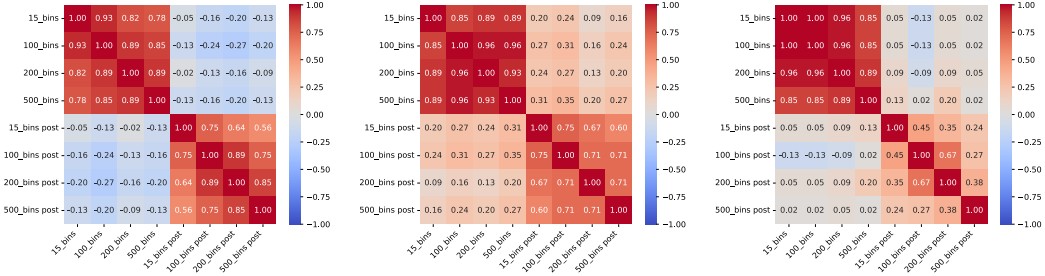

Figure 12: Kendall Ranking Correlation Matrix of ECE before and after temperature scaling on CIFAR-10 (left), CIFAR-100 (middle) and ImageNet (right).

As shown in Figure 12, ECE evaluated on different bin size shows little correlation between pre and post temperature scaling. It indicate that well-calibrated models do not necessarily exhibit better calibration performance after post-hoc calibration techniques.

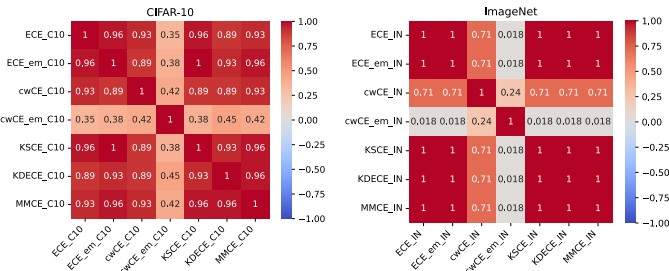

Figure 13: Kendall ranking correlation between diverse calibration metrics. The metrics are evaluated across all transformer models on CIFAR-10(left) and ImageNet(right).

Figure 13 shows the metric correlation among transformer architectures, which shows similar pattern as that on CNNs. The equal mass classwise-CE shows inconsistent results as other metrics. A full correlation map of all metrics before and after temperature scaling are shown in Figure 14 and 15.

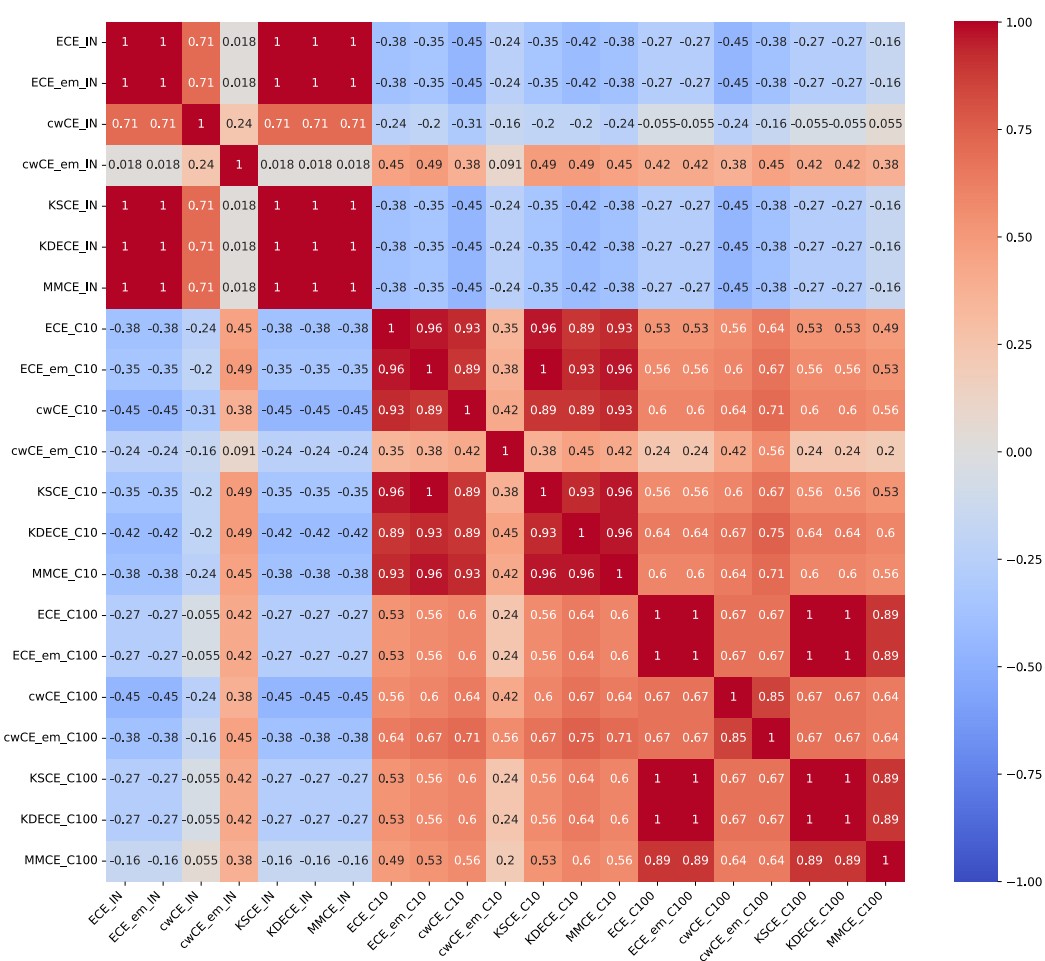

Figure 14: Kendall ranking correlation between all calibration metrics. The metrics are evaluated across all transformer models on CIFAR-10, CIFAR-100 and ImageNet before temperature scaling

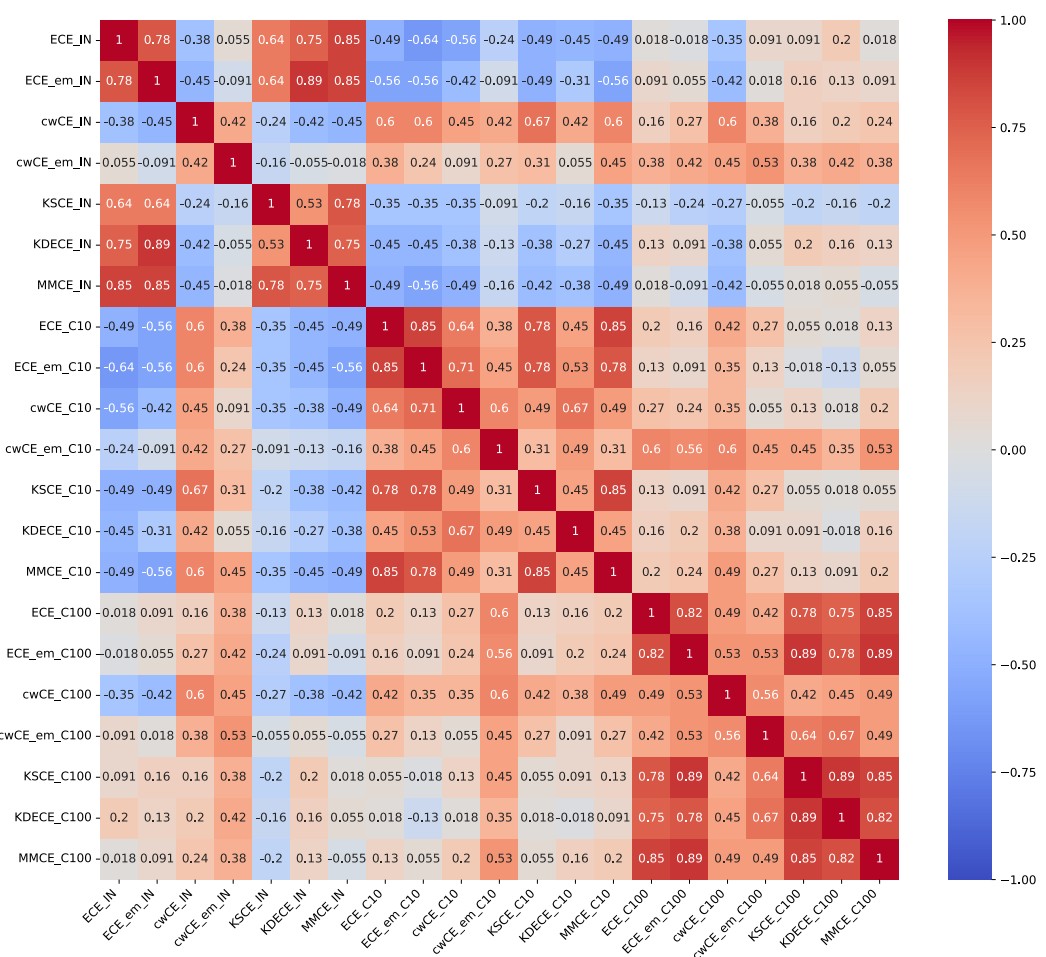

Figure 15: Kendall ranking correlation between all calibration metrics. The metrics are evaluated across all transformer models on CIFAR-10, CIFAR-100 and ImageNet after temperature scaling

## B  EXPERIMENTS ON OTHER CALIBRATION METHODS

To ensure the broad applicability of our findings, we conducted evaluations using 6 human-designed Convolutional Neural Networks (CNNs) with diverse architectures, including ResNet18, ResNet34, ResNet50, ResNet110 (He et al., 2016), Wide-ResNet (Zagoruyko & Komodakis, 2016), and DenseNet121 (Huang et al., 2017). Each CNN underwent 200 epochs of training on CIFAR-10 and CIFAR-100 datasets, employing different loss functions, namely Cross Entropy, Focal Loss (Mukhoti et al., 2020), and MMCE Loss (Kumar et al., 2018). The selected loss functions, Focal Loss and MMCE Loss, are established train-time calibration methods. Our primary objective is to ascertain whether the patterns observed in our study are consistent across various calibration techniques, extending beyond the only calibration approach used in this work, temperature scaling (Guo et al., 2017).

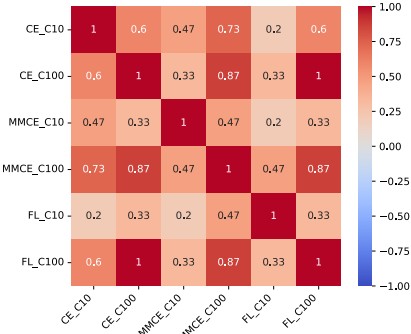

Figure 16: Kendall ranking correlation matrix of ECE for different CNNs.

As depicted in Figure 16, analogous to the observation in Figure 2, models trained using various train-time calibration methods exhibit minimal correlation between their performance on CIFAR-10 and CIFAR-100. This suggests that the calibration characteristics of a specific architecture may not generalize effectively across different datasets.

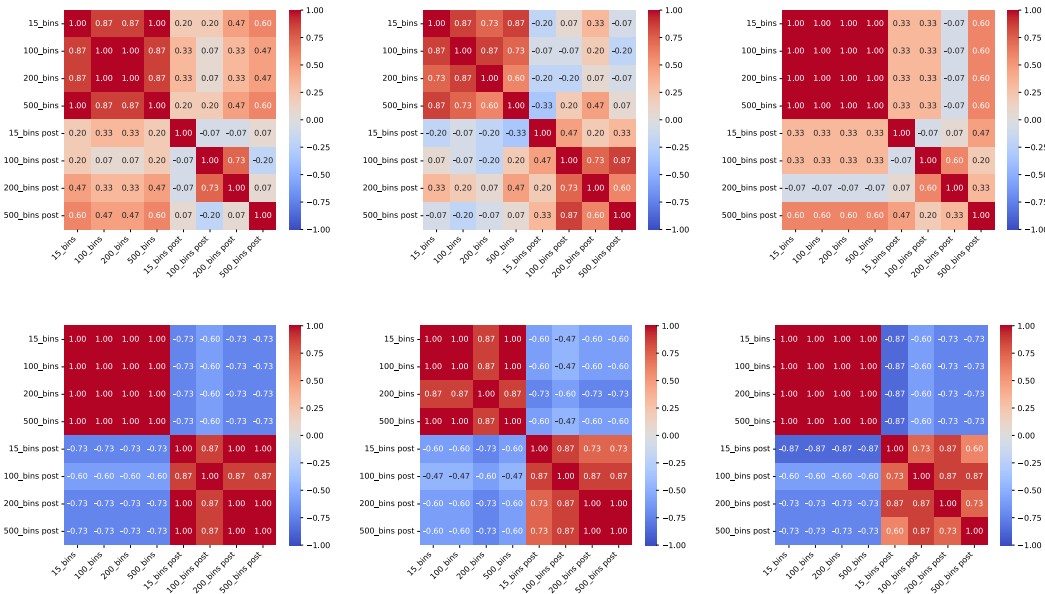

Figure 17: Kendall Ranking Correlation Matrix of ECE before and after temperature scaling on Cross Entropy (left), Focal Loss (middle) and MMCE loss (right). The first row shows results on CIFAR10 and the second row shows results on CIFAR100.

As illustrated in Figure 17, the ECE across different bin sizes demonstrates minimal correlation between pre and post temperature scaling. This suggests that well-calibrated models do not necessarily exhibit enhanced

calibration performance following post-hoc calibration techniques. This trend is particularly pronounced in the case of CIFAR-100, where post-hoc calibration performance is negatively correlated with pre-calibration performance. Notably, the choice of bin size appears to have a more substantial impact on post-hoc calibration performance.

To assess the reliability of calibration metrics, we conducted an analysis of the correlation between all calibration metrics, as presented in Figure 18 and Figure 19. Notably, equal-mass classwise ECE displays a distinct pattern compared to other metrics, particularly on CIFAR-100, reinforcing the observations outlined in Section 4.3.

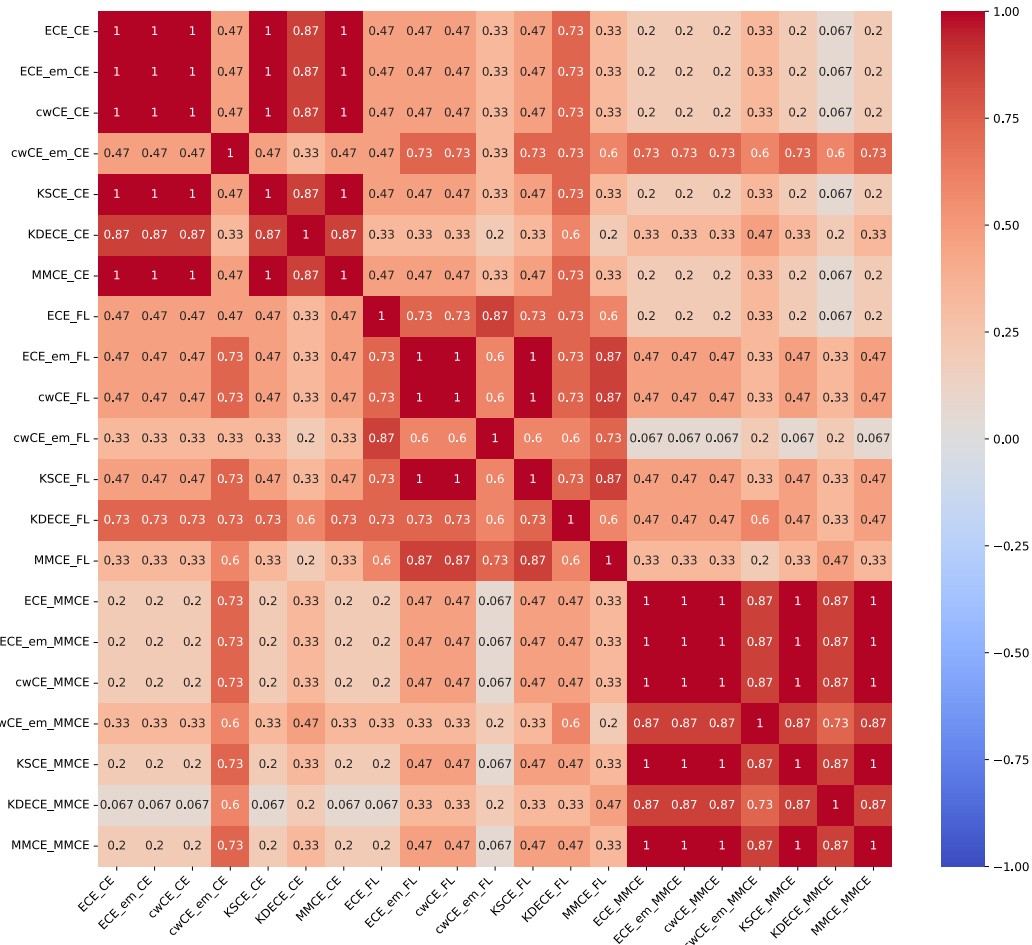

Figure 18: Kendall ranking correlation between all calibration metrics on CIFAR10 for different train time calibration methods.

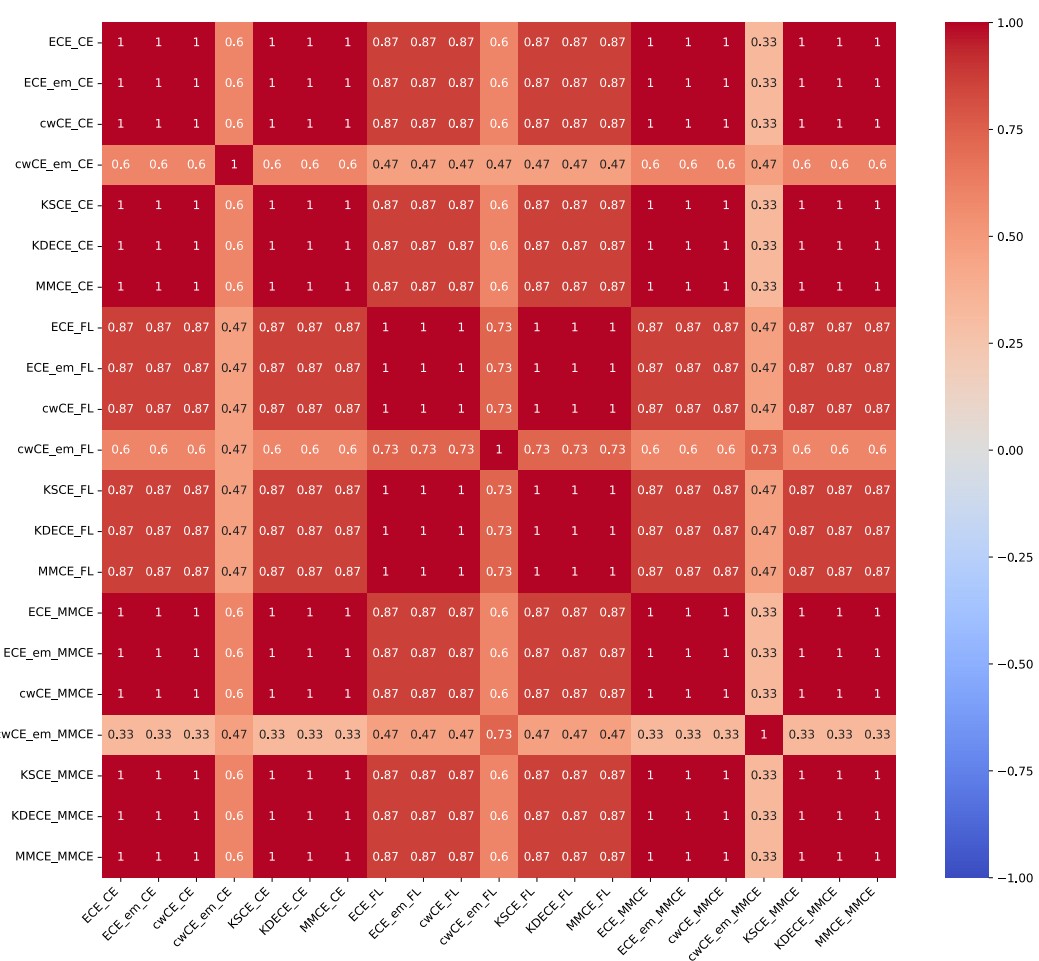

Figure 19: Kendall ranking correlation between all calibration metrics on CIFAR100 for different train time calibration methods.

## C   EXPERIMENTS ON LARGE DATASETS PRETRAINED MODELS

To substantiate our research using modern larger datasets, we assess the performance of seven LAION-2b (Schuhmann et al., 2022) pretrained models on ImageNet-1K (Deng et al., 2009). Additionally, we exam the zero-shot calibration capabilities of large models by evaluating seven pretrained CLIP models on diverse large datasets, including YFCC100M (Thomee et al., 2016) and LAION-400m (Schuhmann et al., 2021), evaluated on CIFAR10 and CIFAR100, respectively.

All pretrained models are downloaded from HuggingFace, seven LAION-5b pretrained models are:

1. vit_large_patch14_clip_224.laion2b_ft_in1k
2. vit_base_patch32_clip_224.laion2b_ft_in1k
3. vit_huge_patch14_clip_224.laion2b_ft_in1k
4. vit_base_patch16_clip_224.laion2b_ft_in1k
5. convnext_large_mlp.clip_laion2b_augreg_ft_in1k
6. convnext_base.clip_laion2b_augreg_ft_in1k
7. vit_base_patch16_clip_224.laion2b_ft_in12k_in1k

Seven zero-shot models are:

1. openai/clip-vit-base-patch16
2. openai/clip-vit-large-patch14
3. openai/clip-vit-base-patch32
4. laion/CLIP-ViT-H-14-laion2B-s32B-b79K
5. patrickjohncyh/fashion-clip
6. flax-community/clip-rsicd-v2
7. flax-community/clip-rsicd

### C.1   PRETRAINED MODELS ON IMAGENET

The right graph of Fig. 20 reveals a pattern consistent with Section 4.4, highlighting that the choice of bin size significantly influences post temperature scaling, while there is minimal correlation between pre temperature scaling ECE and post temperature scaling ECE.

Similarly, in Fig. 21, post temperature scaling calibration metrics exhibit lower correlations. Notably, equal mass classwise ECE demonstrate relatively weaker correlations with others, emphasizing the unreliability of this metric. These observations align with the findings presented in the main paper.

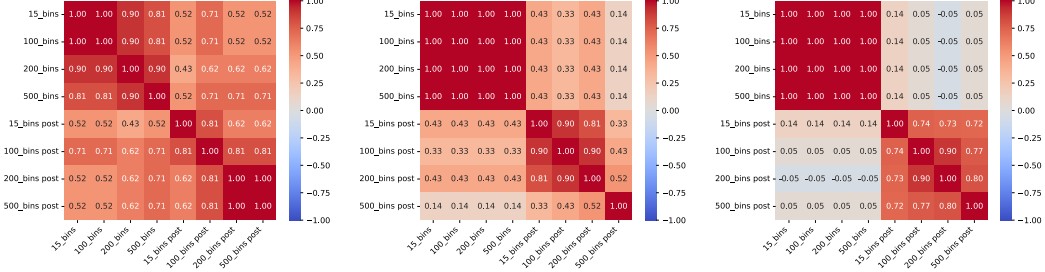

Figure 20: Kendall Ranking Correlation Matrix of ECE before and after temperature scaling with different bin size on CIFAR-10 (left), CIFAR-100 (middle) and ImageNet (right).

### C.2   PRETRAINED MODELS ON ZERO SHOT CALIBRATION

From Fig. 20, a pattern akin to that observed on ImageNet emerges, revealing that bin size exerts a more significant influence on post hoc calibration. Notably, well-calibrated models do not consistently manifest improved calibration performance after post-hoc calibration techniques, as elucidated in section 4.4.

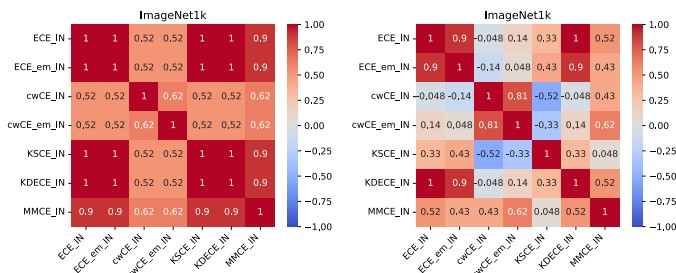

Figure 21: Kendall ranking correlation between diverse calibration metrics on ImageNet before and after temperature scaling.

Fig 22 and Fig 23 further underscore the limited generalizability of calibration properties across diverse datasets, a point thoroughly discussed in section 4.1. This phenomenon is particularly evident in the context of post-temperature scaling calibration error measurements.

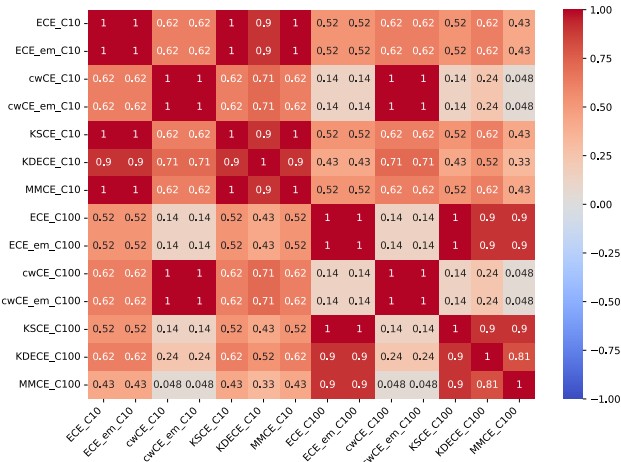

Figure 22: Kendall ranking correlation between all calibration metrics on CIFAR10 and CIFAR100.

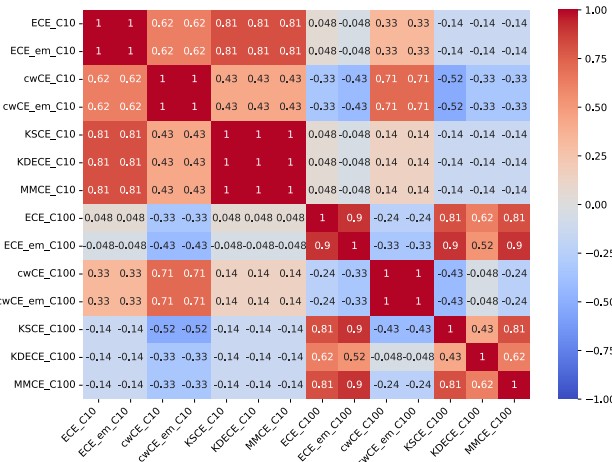

Figure 23: Kendall ranking correlation between all calibration metrics on CIFAR10 and CIFAR100 after temperature scaling.

# D  DATASETS

## D.1  CIFAR-10

This dataset serves as a standard benchmark for image classification tasks, encompassing 60,000 colored images, each of 32x32 pixels, distributed across 10 distinct classes. The original training set houses 50,000 images, allocating 5,000 images for each class, while the test set contains 10,000 images with 1,000 images per class. To facilitate validation, the test images in CIFAR-10 are split into two groups, with 20% allocated for validation purposes.

## D.2  CIFAR-100

Mirroring the structure of CIFAR-10, the CIFAR-100 dataset extends the classification to 100 fine-grained classes while retaining the same image collection. The original training and test sets remain the same with 50,000 and 10,000 images respectively. A random split is performed on the original test set, dividing it into two groups with an 80/20 ratio. The latter group serves as the validation set, while the former constitutes the new test set.

## D.3  IMAGENET16-120

The ImageNet-16-120 dataset is constructed from a down-sampled variant of ImageNet, dubbed ImageNet16×16, which significantly reduces computation costs for optimizing hyperparameters in certain classical models, as indicated in reference (Chrabaszcz et al., 2017). This down-sampling process shrinks the original ImageNet images to 16×16 pixels. From this variant, all images labeled between 1 and 120 are selected to form ImageNet-16-120. The resultant dataset comprises 151.7K training images, 3K validation images, and 3K test images, spread across 120 classes.

# E  REPRODUCIBILITY

Dataset is anonymously available at https://anonymous.4open.science/r/CalibrationDataset5AE6. The README.md file illustrate the use of our dataset. The detailed reproduction process is shown in Algorithm 1.

---

**Algorithm 1** Calibration Metric Dataset Generation

---

    (i) Architecture space $A$ (NATS-Bench)
    (ii) Test datasets $D$ (CIFAR-10, CIFAR-100, ImageNet16-120)
    (iii) Set of calibration metrics $E$
    (iv) Calibration Metric Dataset $C$
 1: **for** $a \in A$ **do**
 2:    ▷ Load pretrained weights for $a$
 3:    $a.load\_weights(d)$
 4:    **for** $d \in D$ **do**
 5:        **for** $eval(\cdot, \cdot) \in E$ **do**
 6:            ▷Evaluate architecture $a$ with $de$
 7:            Calibration errors $\leftarrow eval(a, de)$
 8:            ▷Extend calibration dataset with evaluations
 9:            $C[d][e][\text{“}CE\text{”}][a] \leftarrow$ Calibration errors
10:        **end for**
11:    **end for**
12: **end for**

---

# F NAS ON CALIBRATION

By utilizing the proposed HCS, we are able to evaluate the performance of NAS algorithms in the pursuit of better calibrated models. Specifically, we have selected three representative NAS algorithms - Regularized Evolution (Real et al., 2019), Local Search (White et al., 2021), and Random Search (Li & Talwalkar, 2020) - and have measured their performance in searching models with better calibration performance across different metrics. As indicated in Table 1, Regularized Evolution achieved better balance between prediction accuracy and calibration performance when searching through $HCS_3$.

|  | Method | Acc | ECE | $HCS_1$ | $HCS_2$ | $HCS_3$ | MMCE | KSCE |
|---|---|---|---|---|---|---|---|---|
| Accuracy | Regularized Evolution (Real et al., 2019) | 93.91 | 4.20 | 94.84 | 95.16 | 95.32 | 3.71 | 4.18 |
|  | Local Search (White et al., 2021) | 94.01 | 4.25 | 94.87 | 95.16 | 95.31 | 3.82 | 4.23 |
|  | Random Search (Li & Talwalkar, 2020) | 93.52 | 4.15 | 94.67 | 95.06 | 95.26 | 3.69 | 4.14 |
| $HCS_1$ | Regularized Evolution (Real et al., 2019) | 93.94 | 4.17 | 94.88 | 95.19 | 95.35 | 3.74 | 4.15 |
|  | Local Search (White et al., 2021) | 93.98 | 4.09 | 94.94 | 95.26 | 95.42 | 3.67 | 4.06 |
|  | Random Search (Li & Talwalkar, 2020) | 93.59 | 4.21 | 94.68 | 95.05 | 95.23 | 3.73 | 4.16 |
| $HCS_2$ | Regularized Evolution (Real et al., 2019) | 93.80 | 4.14 | 94.82 | 95.17 | 95.34 | 3.71 | 4.11 |
|  | Local Search (White et al., 2021) | 93.73 | 4.05 | 94.83 | 95.20 | 95.39 | 3.62 | 4.01 |
|  | Random Search (Li & Talwalkar, 2020) | 93.06 | 4.12 | 94.45 | 94.92 | 95.16 | 3.58 | 4.08 |
| $HCS_3$ | Regularized Evolution (Real et al., 2019) | 93.62 | 3.90 | 94.84 | 95.26 | 95.47 | 3.42 | 3.86 |
|  | Local Search (White et al., 2021) | 93.57 | 3.91 | 94.81 | 95.24 | 95.45 | 3.42 | 3.87 |
|  | Random Search (Li & Talwalkar, 2020) | 93.55 | 4.23 | 94.65 | 95.02 | 95.21 | 3.71 | 4.17 |

Table 1: **NAS methods search on HCS and Accuracy**

# G  BOX-PLOT OF ECE OF DIFFERENT BIN SIZE IN TSS

In this section, we provide the box plots on ECE with different bin sizes for all models in TSS at different epochs, both before and after temperature scaling, on each dataset.

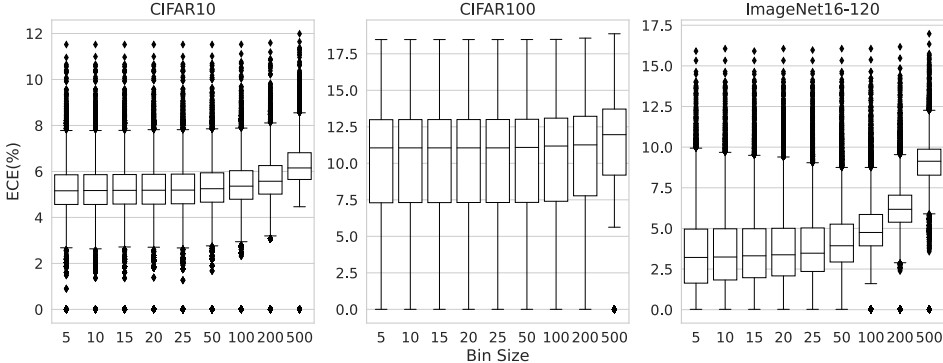

Figure 24: Box plots of ECE with different bin sizes are shown for models trained with 200 epochs in TSS before temperature scaling.

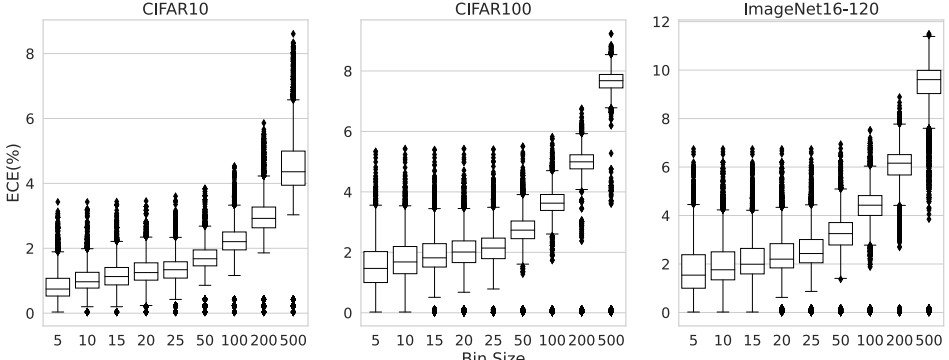

Figure 25: Box plots of ECE with different bin sizes are shown for models trained with 200 epochs in TSS after temperature scaling.

# H  BOX-PLOT OF ECE OF DIFFERENT BIN SIZE IN SSS

In this section, we provide the box plots of ECE with different bin sizes for all models in SSS at 90 epochs, both before and after temperature scaling, on each dataset.

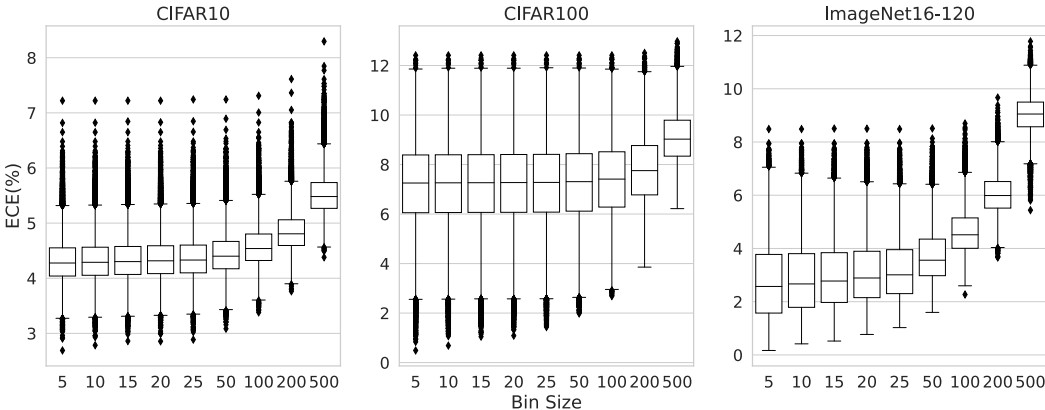

Figure 26: Box plots of ECE with different bin sizes are shown for models trained with 90 epochs in SSS before temperature scaling.

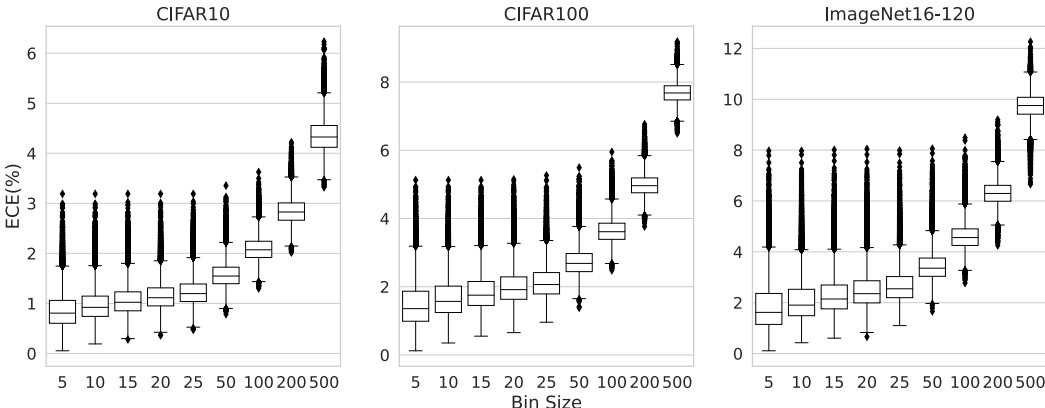

Figure 27: Box plots of ECE with different bin sizes are shown for models trained with 90 epochs in SSS after temperature scaling.

# I BOX-PLOT OF ECE AND ACCURACY WITH MODEL SIZE

In this section, we present the box plots of ECE and Accuracy across all architectures within the SSS subset, segregated by different size brackets. The model size is defined as the total number of kernels in each layer.

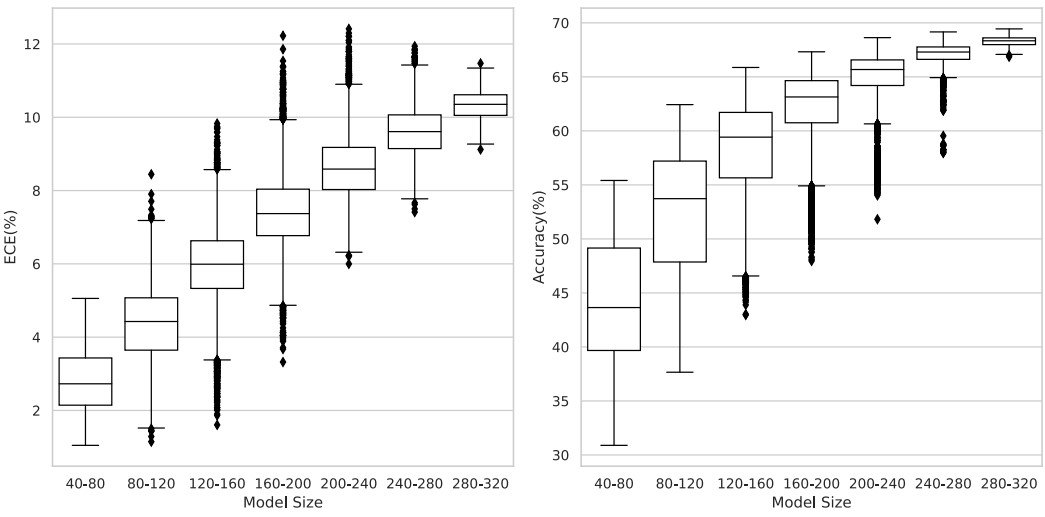

Figure 28: Box-plots measured on CIFAR-100 for ECE and Accuracy for all models in SSS.

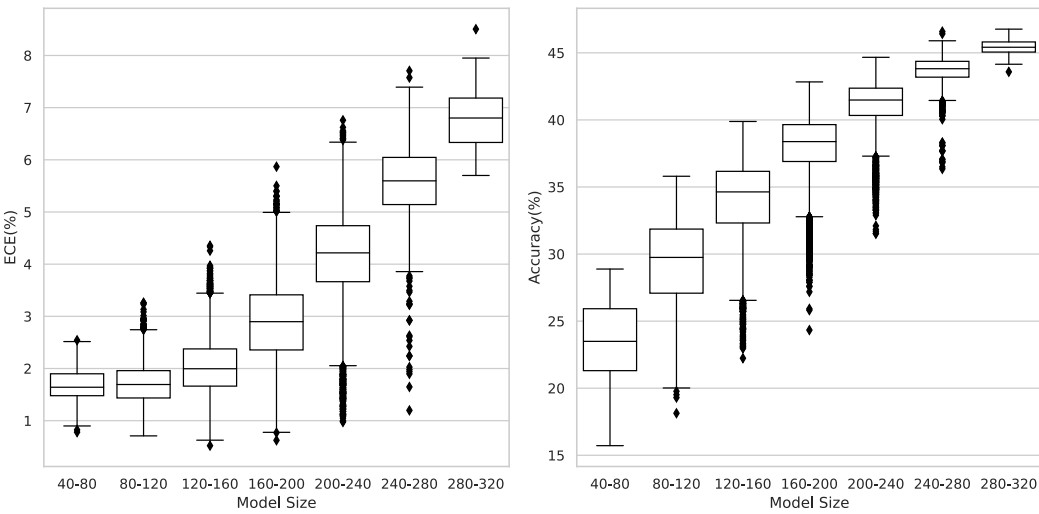

Figure 29: Box-plots measured on ImageNet16-120 for ECE and Accuracy for all models in SSS.

## J  CALIBRATION METRICS

In our study, we consider five bin-based metrics, including Expected Calibration Error (ECE), Equal Mass Binning calibration error (ECE$em$), Classwise Equal-Width Binning calibration error (cwCE), Classwise Equal-Mass Binning calibration error (cwCE$em$), and Maximum Calibration Error (MCE). We evaluate these metrics across a wide range of bin sizes, such as 5, 10, 15, 20, 25, 50, 100, 200, and 500 bins. These metrics are assessed both before and after the application of temperature scaling. In addition to these, we examine three statistics-based metrics: Kolmogorov-Smirnov calibration error (KSCE), kernel-based Top-Label calibration error using Kernel Density Estimation (KDECE), and Maximum Mean calibration error (MMCE). We also consider two measures related to training loss: Negative Log-Likelihood (NLL) and Brier score. Furthermore, we compute the Area Under the Curve (AUC) for the True Skill Statistic (TSS) for models trained on the CIFAR-10 dataset, using two Out-of-Distribution (OoD) datasets, CIFAR-10-C (corrupted with Gaussian noise) and SVHN (Street View House Numbers). Overall, this results in the assessment of 102 different measures, providing a comprehensive evaluation of model calibration.

**Bin-based Metrics** The Top-Label Equal-Width Binning, also known as ECE (Naeini et al., 2015; Guo et al., 2017), partitions the probability interval [0, 1] into equal-sized subintervals. Each data point is represented by the maximum probability output of the predicted class, dividing predictions into $m$ bins $B_1, \ldots, B_m$. Confidence and accuracy for each bin $B_i$ are calculated as $\text{confidence}(B_i) = \frac{1}{|B_i|} \sum_{j \in B_i} \max f(x_j)$ and $\text{accuracy}(B_i) = \frac{1}{|B_i|} \sum_{j \in B_i} \nVdash y_j \in \arg\max f(x_j)$, with $\nVdash \cdot$ being the Iverson bracket. The ECE is computed as $\text{ECE} = \sum_{i=1}^{m} \frac{|B_i|}{n} |\text{accuracy}(B_i) - \text{confidence}(B_i)|$. In contrast, Equal Mass Binning calibration error (ECE$_{em}$) (Kumar et al., 2019; Nixon et al., 2019) uses an adaptive binning scheme, adjusting bin intervals to contain an equal number of predictions, addressing the bias-variance trade-off. Class-wise calibration errors have been proposed in (Kumar et al., 2019; Nixon et al., 2019; Kull et al., 2019), including Classwise Equal-Width Binning calibration error (cwCE) and Classwise Equal-Mass Binning calibration error (cwCE$_{em}$). The cwCE method bins predictions per class, computes the calibration error within the bin, and averages across bins. Confidence and accuracy for each class label $k$ and bin $B_b$ are calculated as $\text{confidence}(B_b, k) = \frac{1}{|B_{b,k}|} \sum_{j \in B_{b,k}} f(x_j)$ and $\text{accuracy}(B_b, k) = \frac{1}{|B_{b,k}|} \sum_{j \in B_{b,k}} \nVdash y_j \in \arg\max f(x_j)$. The cwCE is then computed as $\text{cwCE} = \frac{1}{K} \sum_{k=1}^{K} \sum_{b=1}^{B} \frac{n_{b,k}}{N} |\text{accuracy}(B_b, k) - \text{confidence}(B_b, k)|$. cwCE$_{em}$ applies Equal-Mass binning to the multi-class setting, ensuring each bin contains an equal number of predictions for each class, offering an adaptive approach to multi-class calibration. Maximum Calibration Error (MCE) represents the worst-case error in the model's calibration and measures the maximum difference between confidence and accuracy over all bins (Naeini et al., 2015). It focuses on the most miscalibrated predictions, providing a robustness check for the calibration performance of models.

**Other Metrics** Brier Score (BS): For a model $f : X \rightarrow P_n$, the Brier score is defined as:

$$BS(f) = E\left[||f(X) - Y'||_2^2\right], \tag{1}$$

where $Y'$ is the one-hot encoded version of $Y$. The Brier score is equivalent to the mean squared error, which suggests it captures more than just model calibration; it also measures model fit.

Lp Calibration Error (CEp): For $1 \leq p \leq 2 \in \mathbb{R}$, the Lp calibration error of model $f : X \rightarrow P_n$ is defined as:

$$CE_p(f) = \left(E\left[||f(X) - P(Y|f(X))||_p^p\right]\right)^{\frac{1}{p}}. \tag{2}$$

Kernel Density Estimation (KDE) based ECE: Let $K : \mathbb{R} \rightarrow \mathbb{R}_{\geq 0}$ denote a smoothing kernel function. Given a fixed bandwidth $h > 0$, we have $K_h(a) = h^{-1} K(a/h)$. The unknown probabilities are estimated using KDE as follows:

$$\tilde{p}(z) = \frac{1}{h^L n_e} \sum_{i=1}^{n_e} \prod_{l=1}^{L} K_h(z_l - z_l^{(i)}), \tag{3}$$

$$\tilde{\pi}(z) = \frac{\sum_{i=1}^{n_e} y^{(i)} \prod_{l=1}^{L} K_h(z_l - z_l^{(i)})}{\sum_{i=1}^{n_e} \prod_{l=1}^{L} K_h(z_l - z_l^{(i)})}, \tag{4}$$

$$ECE_{gd}(f) = \int |z - \tilde{\pi}(z)|^d \, d\tilde{p}(z). \tag{5}$$

Kolmogorov-Smirnov calibration error (KS): This measures the maximum deviation between the predicted and actual probabilities. Given model $f : X \rightarrow P_n$, where $C = \arg\max_k f_k(X)$ and $KS(f, k)$ represents the absolute difference between the actual and predicted probabilities for class $k$, the Kolmogorov-Smirnov calibration error is defined as:

$$KS(f) = E[KS(f, C)] = \sup_{z \in [0,1]} \left| \int_0^z P(Y = k | f_k(X) \leq z) dP_{f_k(X)}(z) \right|. \tag{6}$$

Maximum Mean Calibration Error (MMCE): Given a reproducing kernel Hilbert space $H$ with kernel $k : [0, 1] \times [0, 1] \to \mathbb{R}$, the maximum mean calibration error of model $f : X \to P_n$ is defined as:

$$\text{MMCE}(f) = \sup_{h \in H, ||h|| \leq 1} |E\left[(f_C(X) - P(Y = C|f_C(X))) \, k\left(f_C(X), .\right)\right]| . \tag{7}$$

# K  CALIBRATION KENDALL RANKING CONFUSION MATRIX FOR TSS

In Figure 2, we plot a Kendall correlation matrix of ECE for different datasets. In Figure 4a, we plot a Kendall correlation matrix for different metrics. In this section, we provide comprehensive correlation matrix between calibration metrics on different datasets on TSS, filtered by different top accuracy populations.

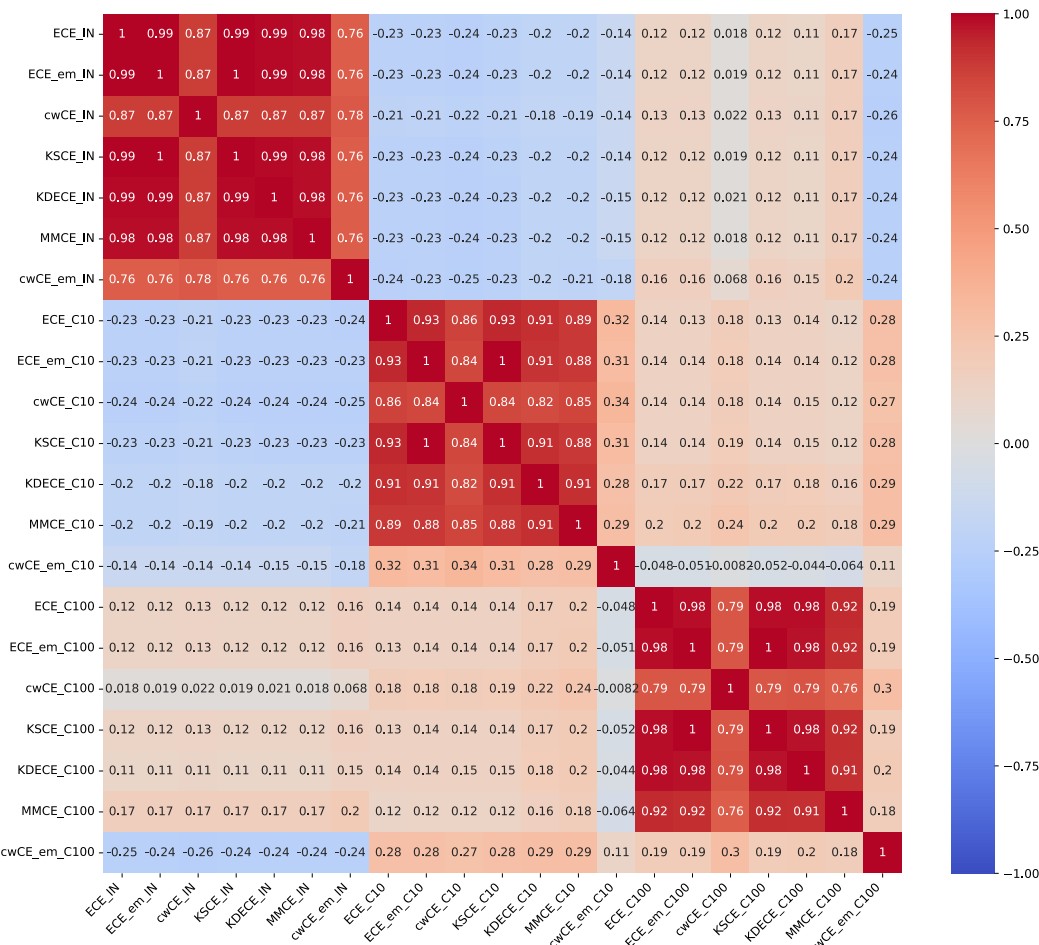

Figure 30: Kendall Ranking Correlation Matrix for the CIFAR-10, CIFAR-100, and the ImageNet16-120 dataset with calibration metrics measured on TSS, filtered by top 100 accuracy.

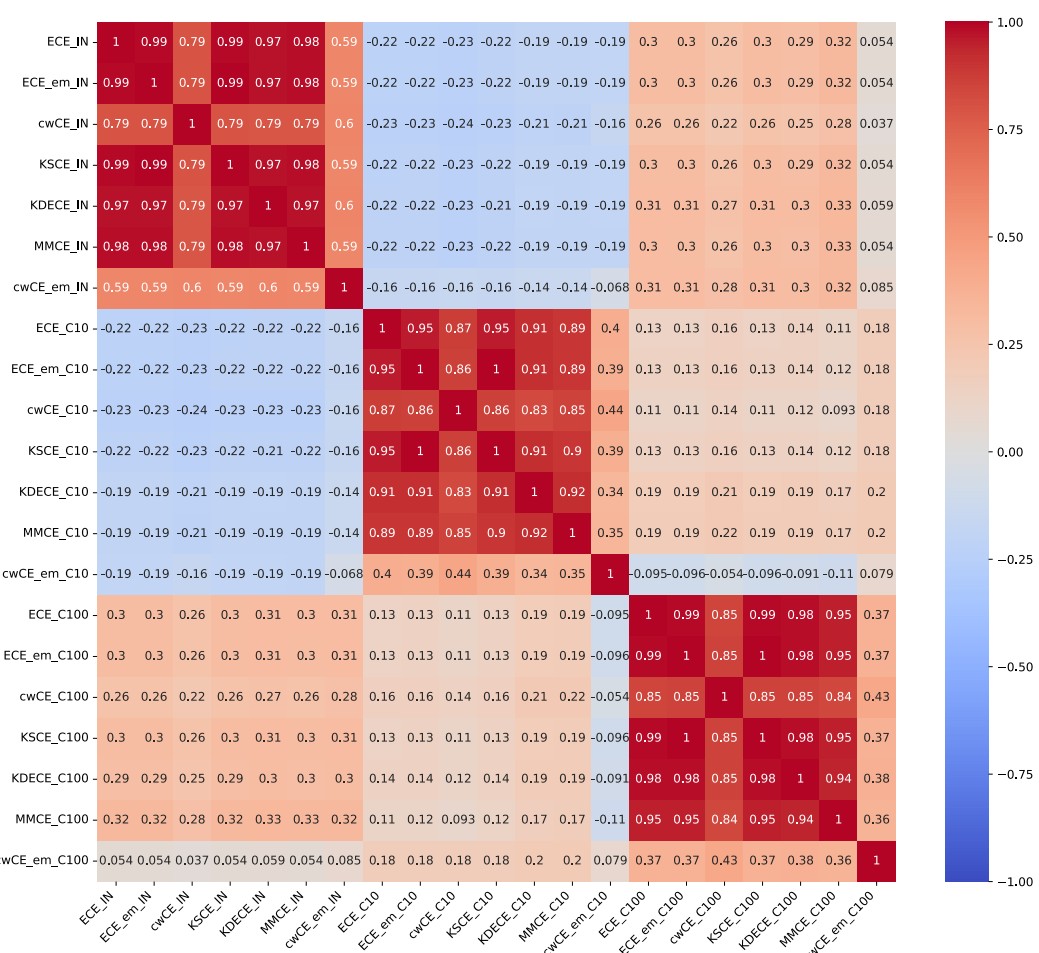

Figure 31: Kendall Ranking Correlation Matrix for the CIFAR-10, CIFAR-100, and the ImageNet16-120 dataset with calibration metrics measured on TSS, filtered by top 1000 accuracy.

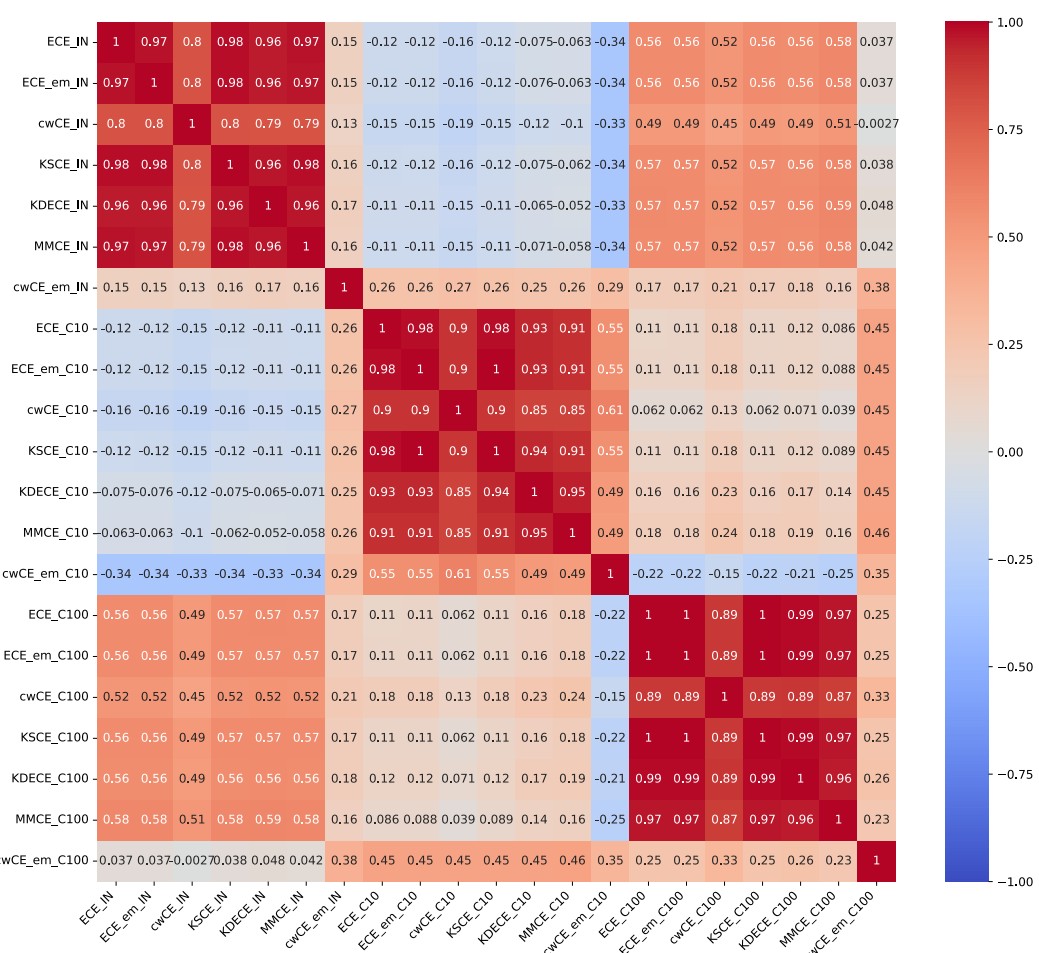

Figure 32: Kendall Ranking Correlation Matrix for the CIFAR-10, CIFAR-100, and the ImageNet16-120 dataset with calibration metrics measured on TSS, filtered by top 5000 accuracy.

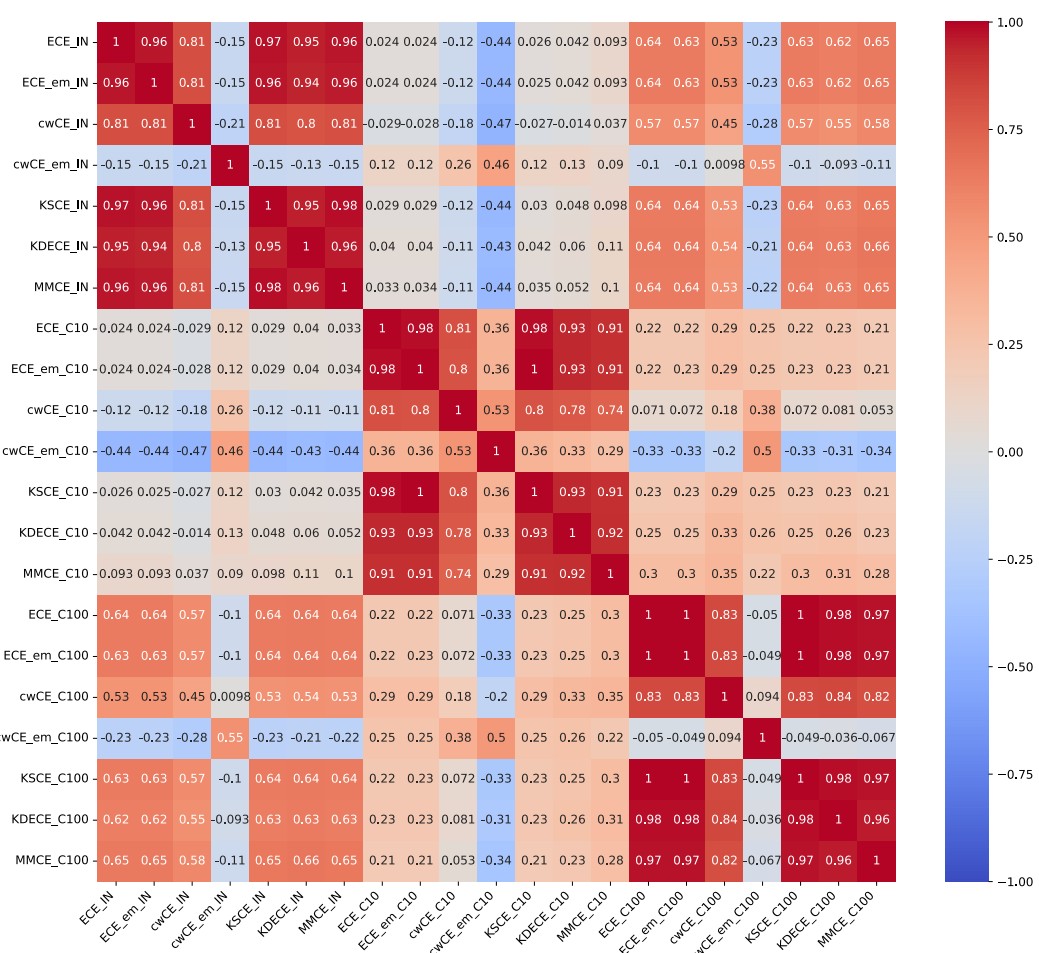

Figure 33: Kendall Ranking Correlation Matrix for the CIFAR-10, CIFAR-100, and the ImageNet16-120 dataset with calibration metrics measured on all models in TSS.

## L Calibration Kendall Ranking Confusion Matrix for SSS

In this section, we provide comprehensive correlation matrix between calibration metrics on different datasets on SSS, filtered by different top accuracy populations.

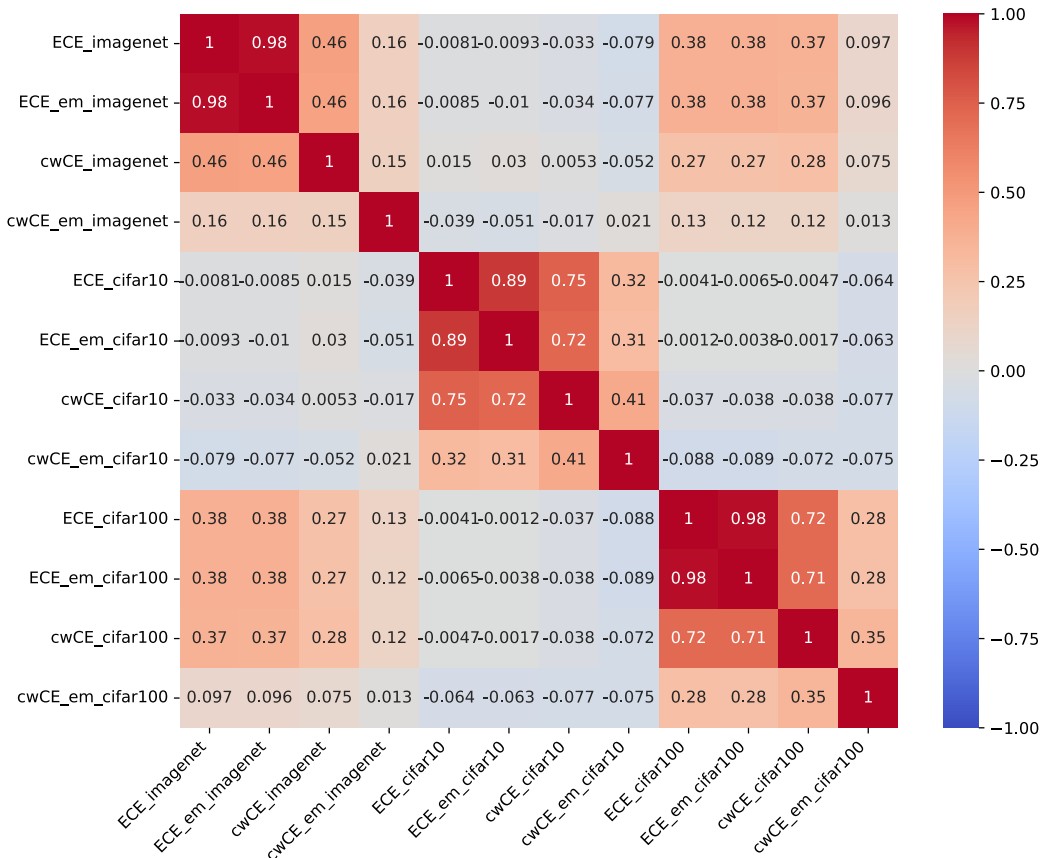

Figure 34: Kendall Ranking Correlation Matrix for the CIFAR-10, CIFAR-100, and the ImageNet16-120 dataset with calibration metrics measured on SSS, filtered by top 100 accuracy.

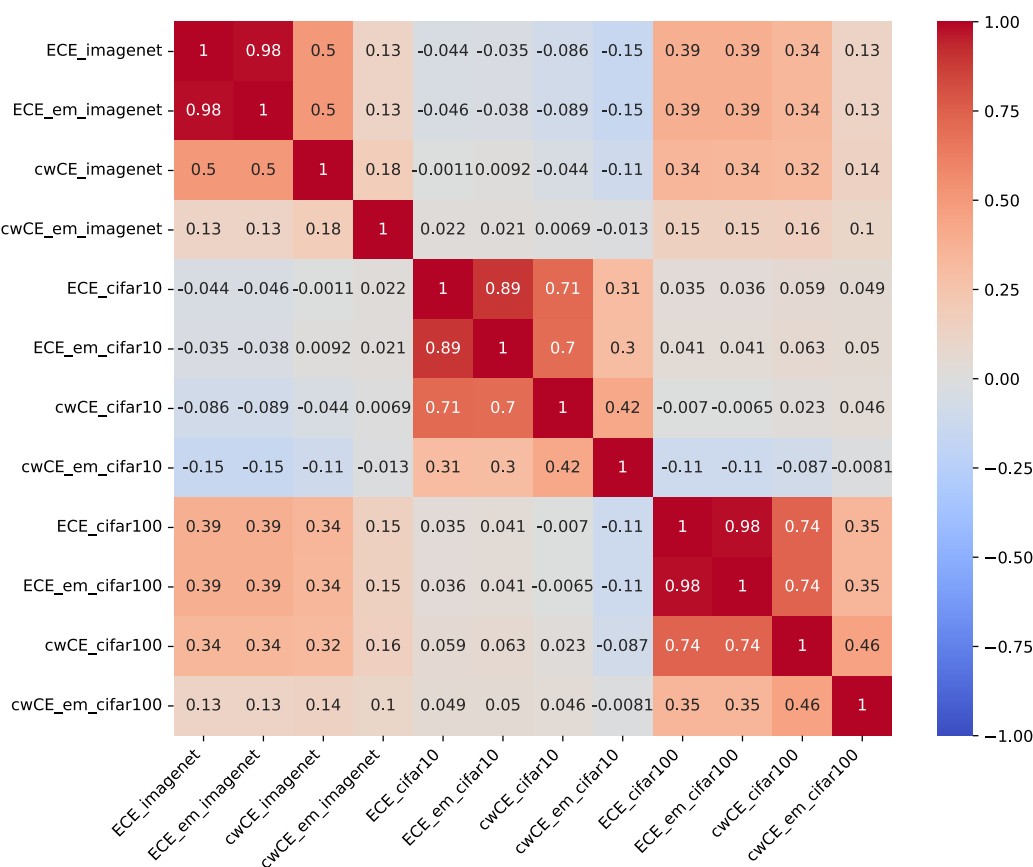

Figure 35: Kendall Ranking Correlation Matrix for the CIFAR-10, CIFAR-100, and the ImageNet16-120 dataset with calibration metrics measured on SSS, filtered by top 1000 accuracy.

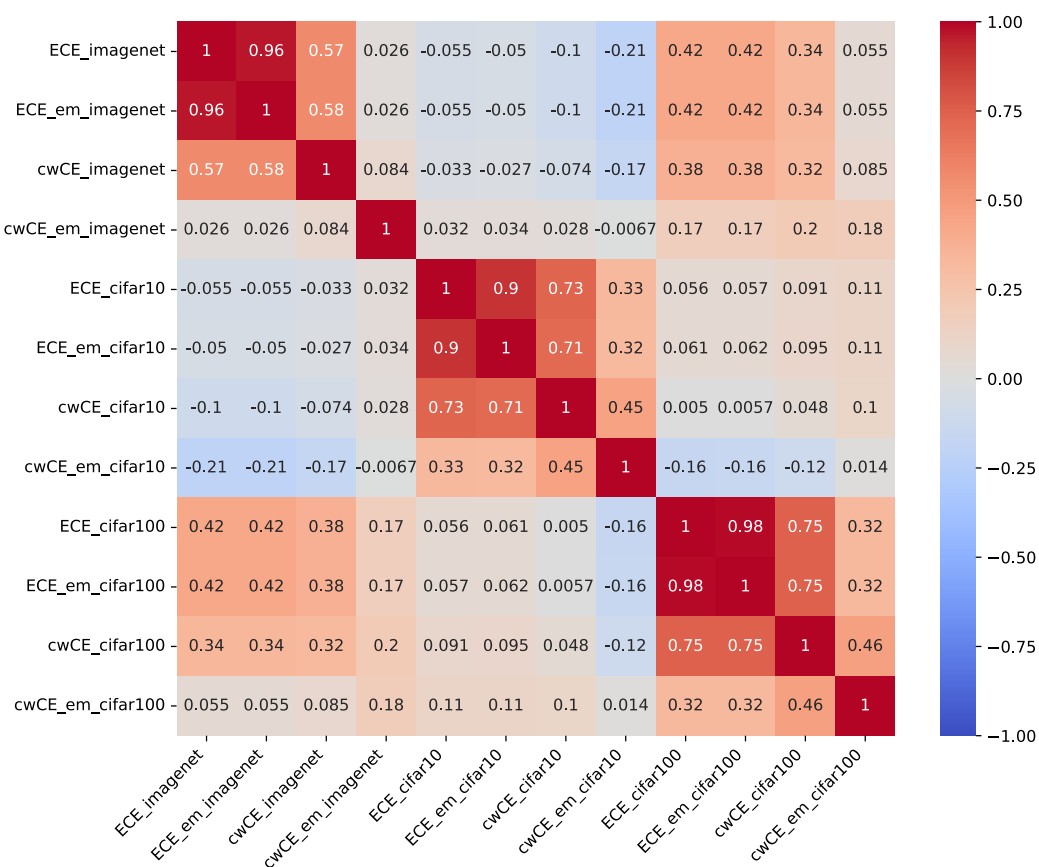

Figure 36: Kendall Ranking Correlation Matrix for the CIFAR-10, CIFAR-100, and the ImageNet16-120 dataset with calibration metrics measured on SSS, filtered by top 5000 accuracy.

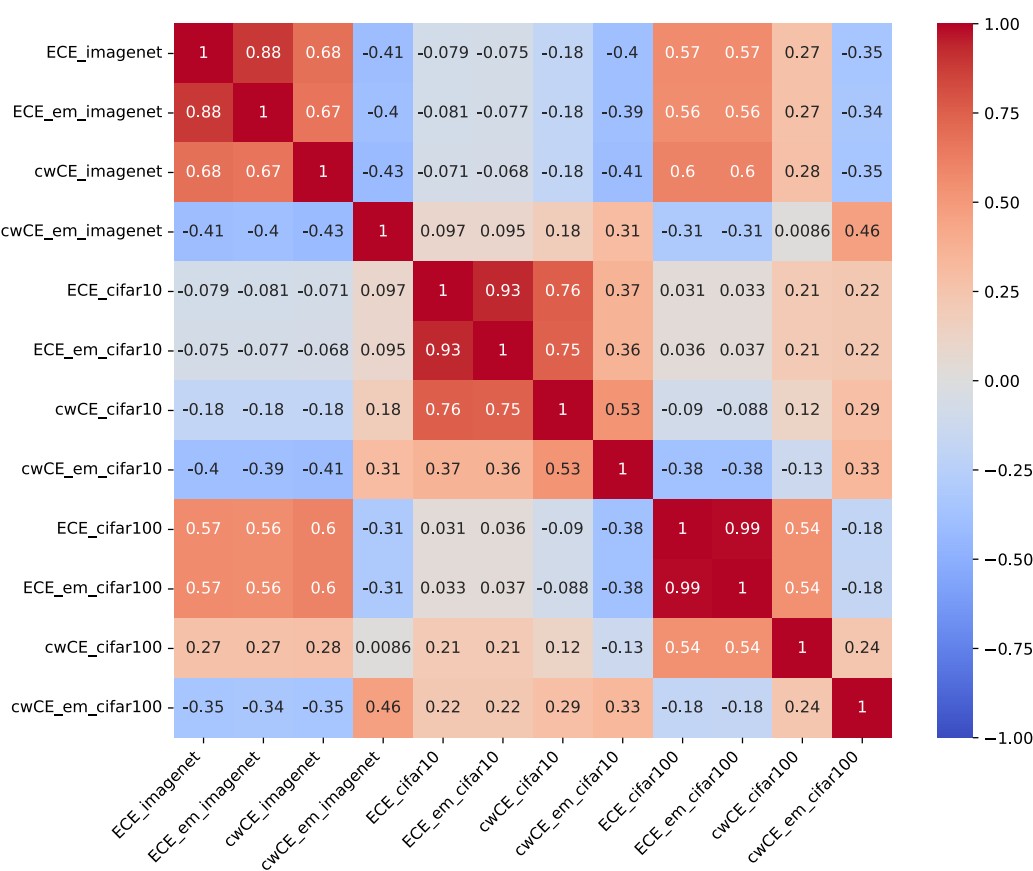

Figure 37: Kendall Ranking Correlation Matrix for the CIFAR-10, CIFAR-100, and the ImageNet16-120 dataset with calibration metrics measured on all models in SSS.

# M    KENDALL RANKING CORRELATION MATRIX BEFORE AND AFTER TEMPERATURE SCALING

In this section, we provide the full correlation matrix between ECE evaluated with different bin size and pre- and post-temperature scaling.

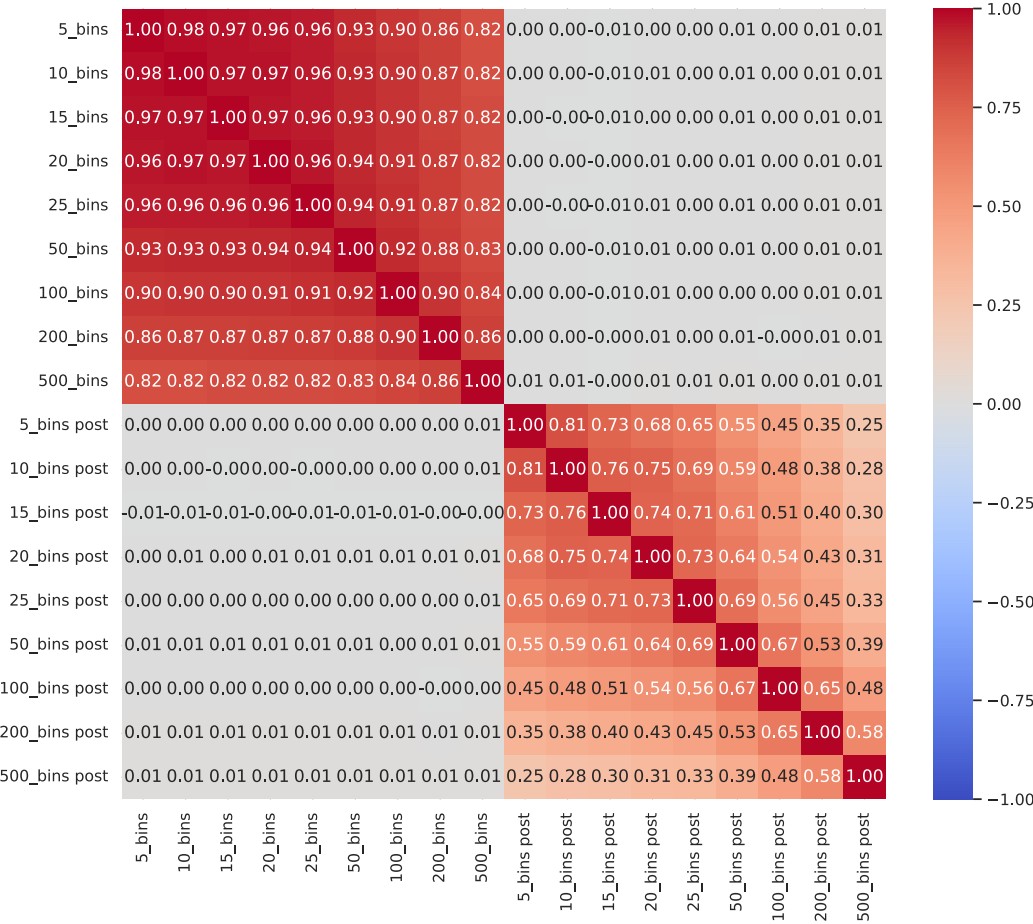

Figure 38: Kendall Ranking Correlation Matrix of ECE using different bin size before and after temperature scaling on CIFAR-10.

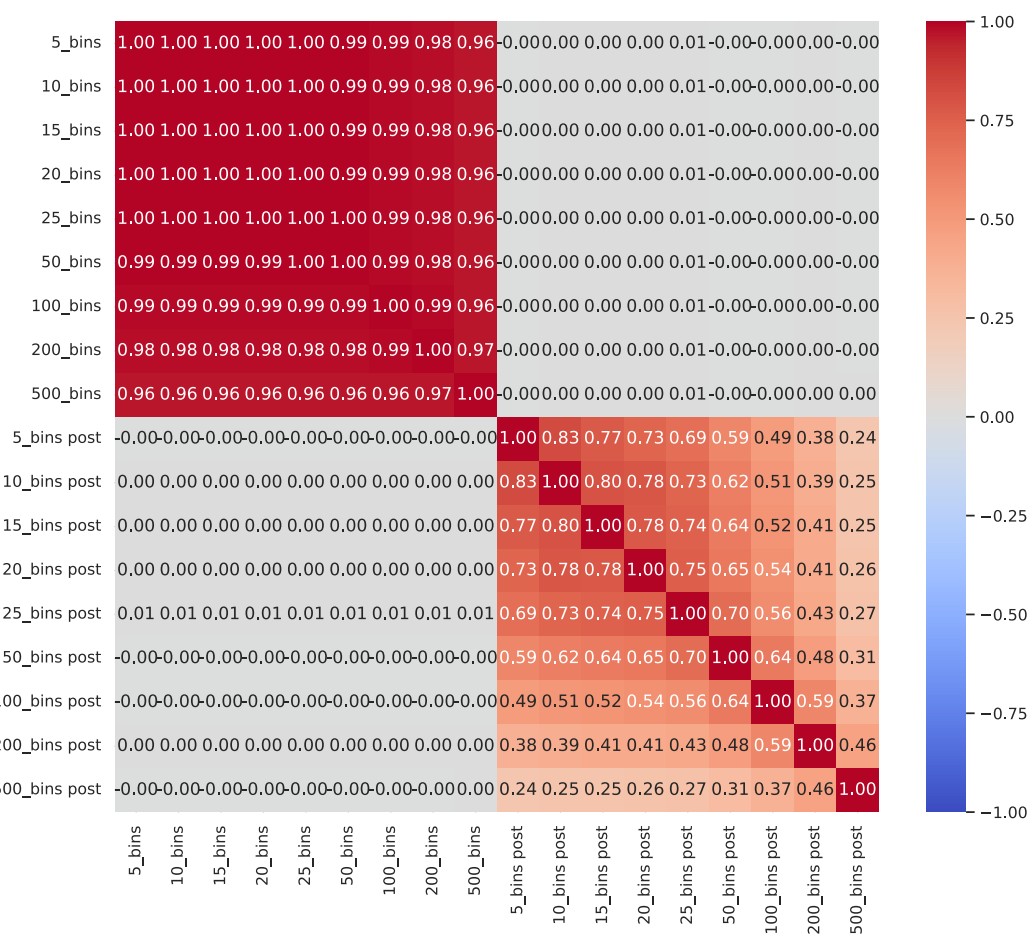

Figure 39: Kendall Ranking Correlation Matrix of ECE using different bin size before and after temperature scaling on CIFAR-100.

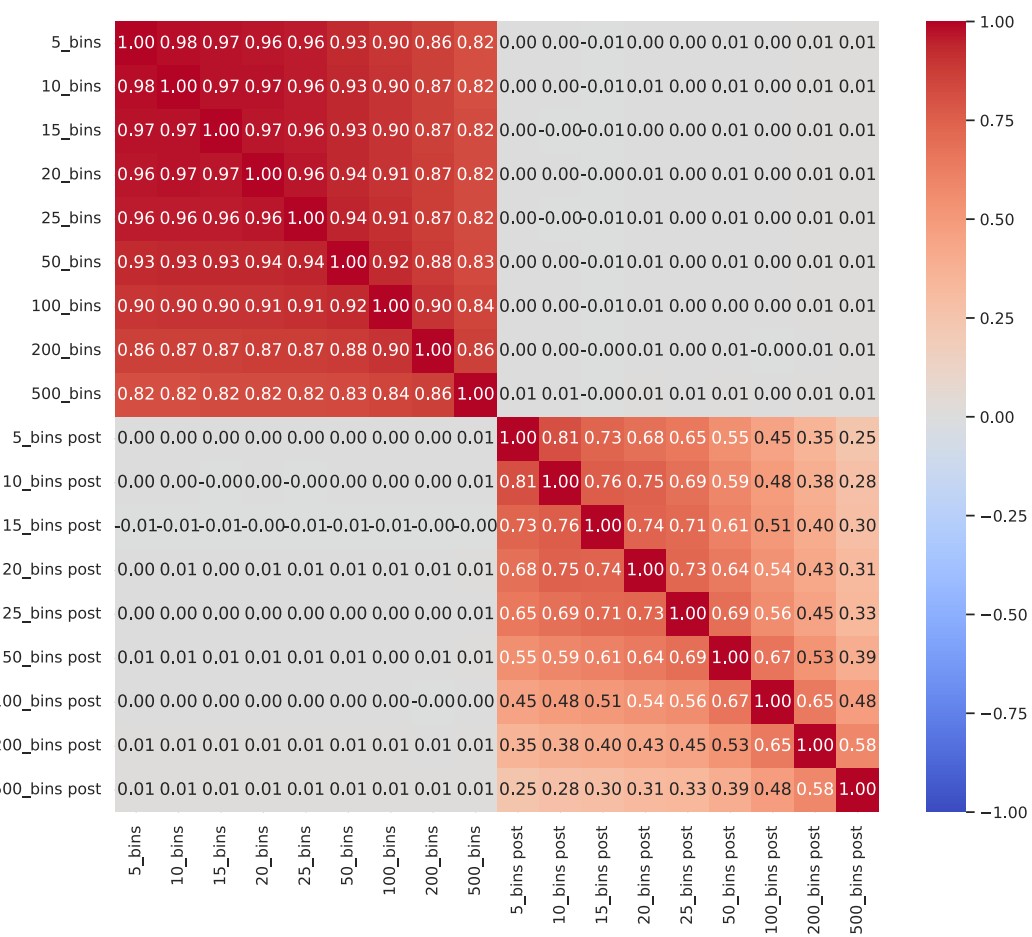

Figure 40: Kendall Ranking Correlation Matrix of ECE using different bin size before and after temperature scaling on ImageNet16-120.

# N RELATIONSHIP ACCURACY, ROBUSTNESS AND CALIBRATION MEASUREMENTS

In Figure 3, we plot a correlation coefficient line for selected metrics. In this section, we provide correlation matrix between key robustness metrics and calibration metrics on different datasets along multiple top ranked populations.

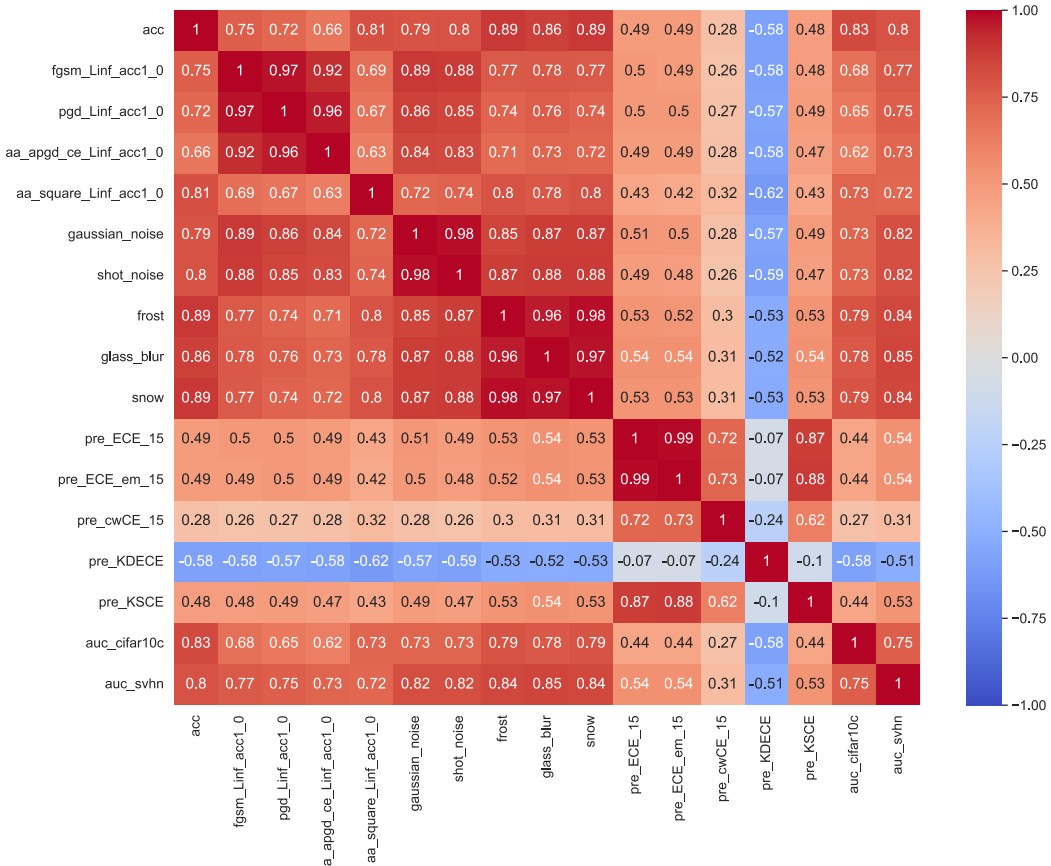

Figure 41: Kendall Ranking Correlation Matrix for the CIFAR-10 with calibration metrics measured on TSS, filtered by top 1000 accuracy. The Correlation Matrix presents the ranking correlations between 5 different calibration metrics and various adversarial attack and image perturbation robustness measurements.

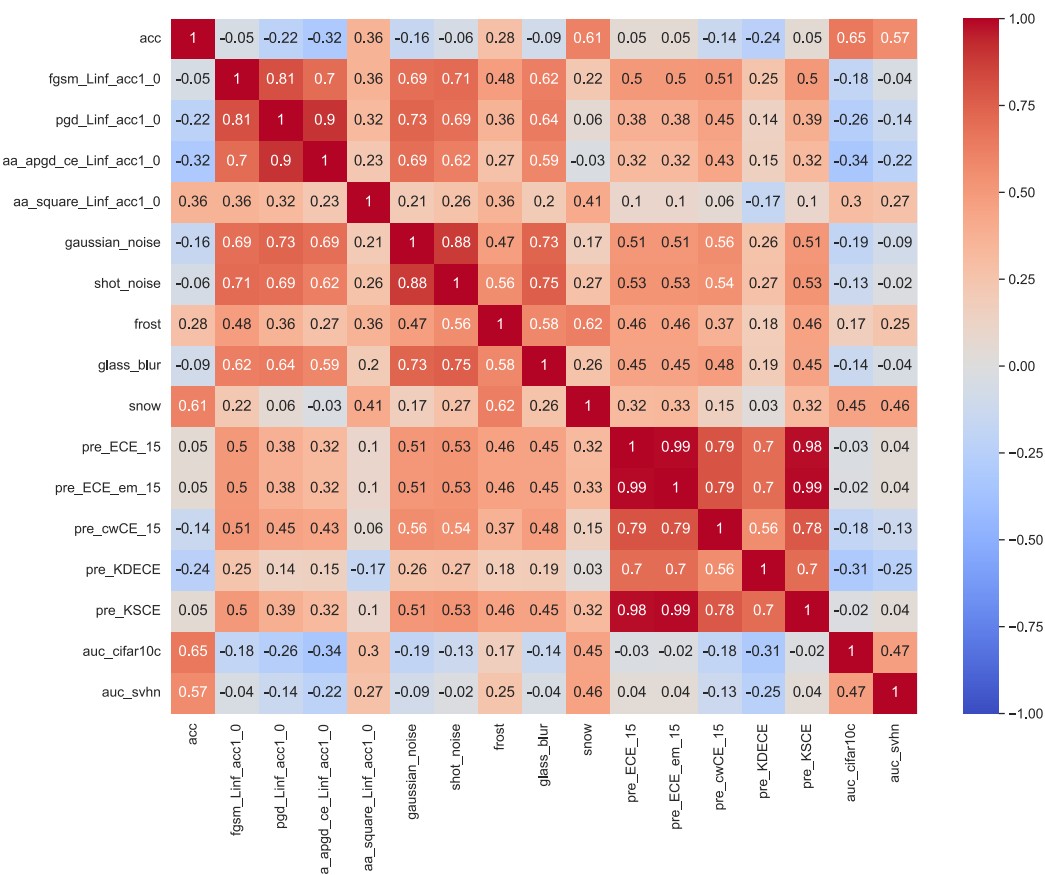

Figure 42: Kendall Ranking Correlation Matrix for the CIFAR-10 with calibration metrics measured on TSS, filtered by top 5000 accuracy. The Correlation Matrix presents the ranking correlations between 5 different calibration metrics and various adversarial attack and image perturbation robustness measurements.

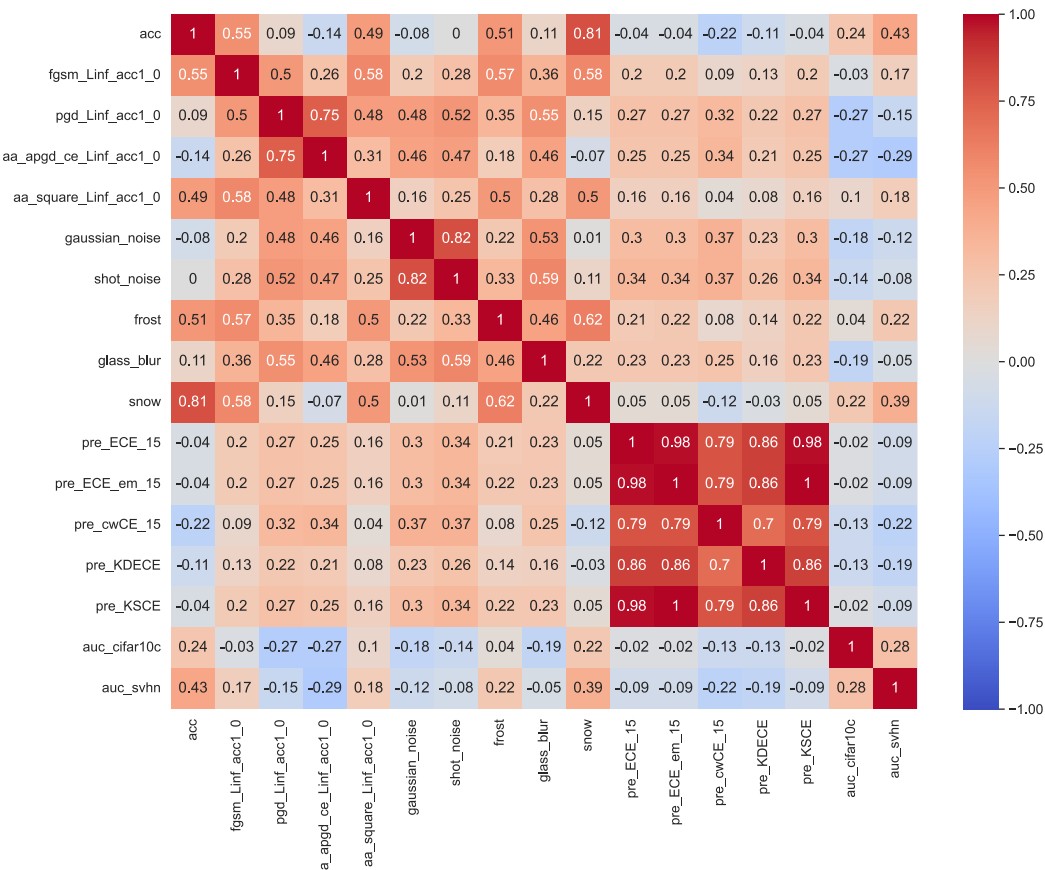

Figure 43: Kendall Ranking Correlation Matrix for the CIFAR-10 with calibration metrics measured on all models in TSS. The Correlation Matrix presents the ranking correlations between 5 different calibration metrics and various adversarial attack and image perturbation robustness measurements.

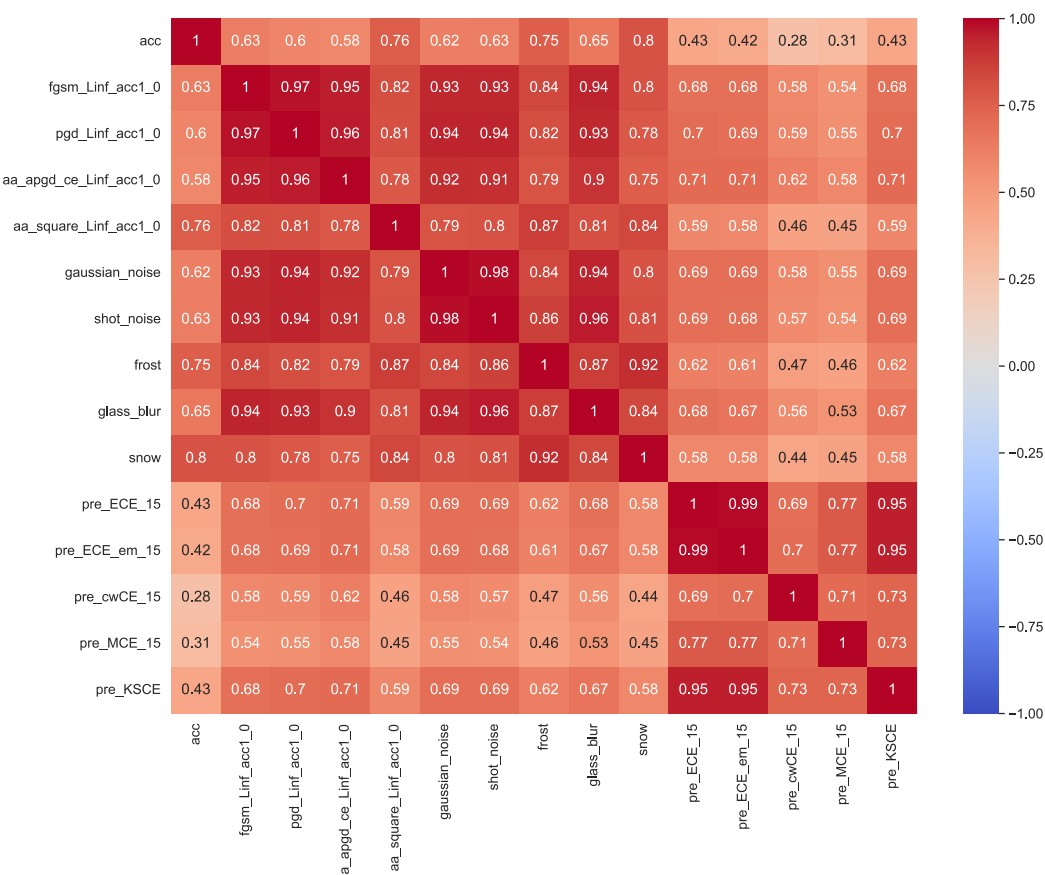

Figure 44: Kendall Ranking Correlation Matrix for the CIFAR-100 with calibration metrics measured on TSS, filtered by top 1000 accuracy. The Correlation Matrix presents the ranking correlations between 5 different calibration metrics and various adversarial attack and image perturbation robustness measurements.

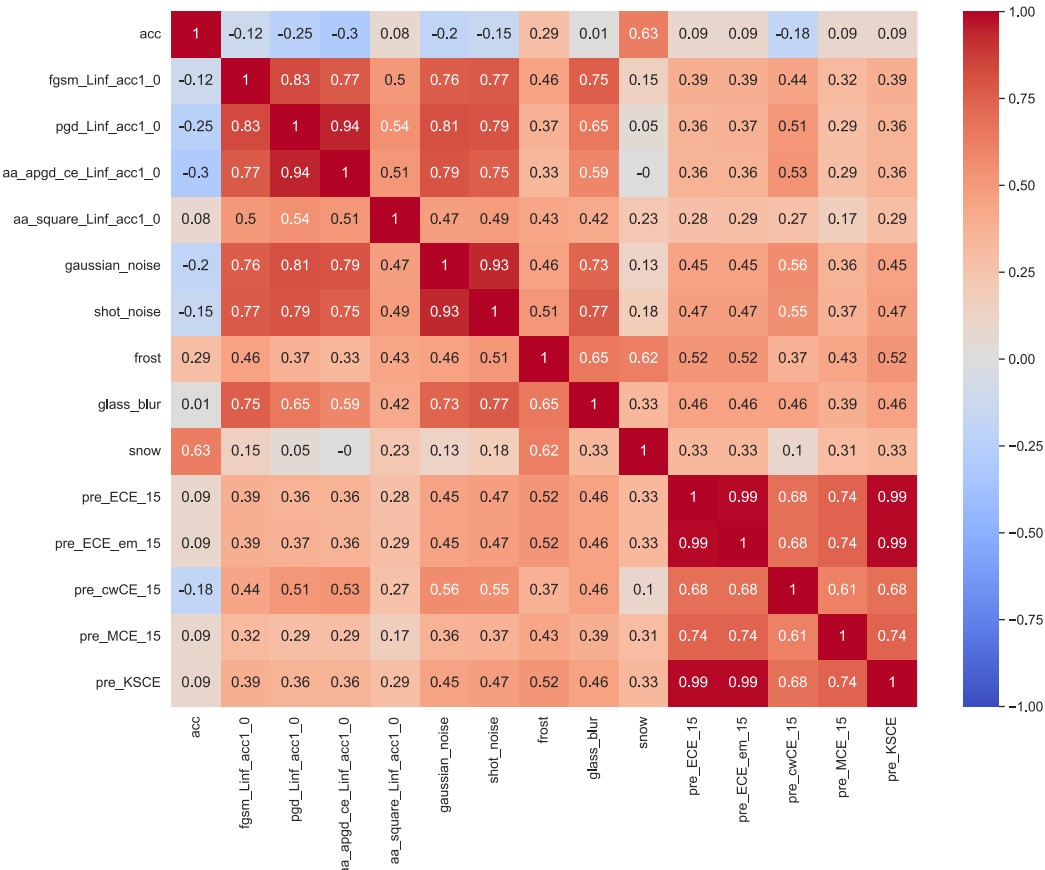

Figure 45: Kendall Ranking Correlation Matrix for the CIFAR-100 with calibration metrics measured on TSS, filtered by top 5000 accuracy. The Correlation Matrix presents the ranking correlations between 5 different calibration metrics and various adversarial attack and image perturbation robustness measurements.

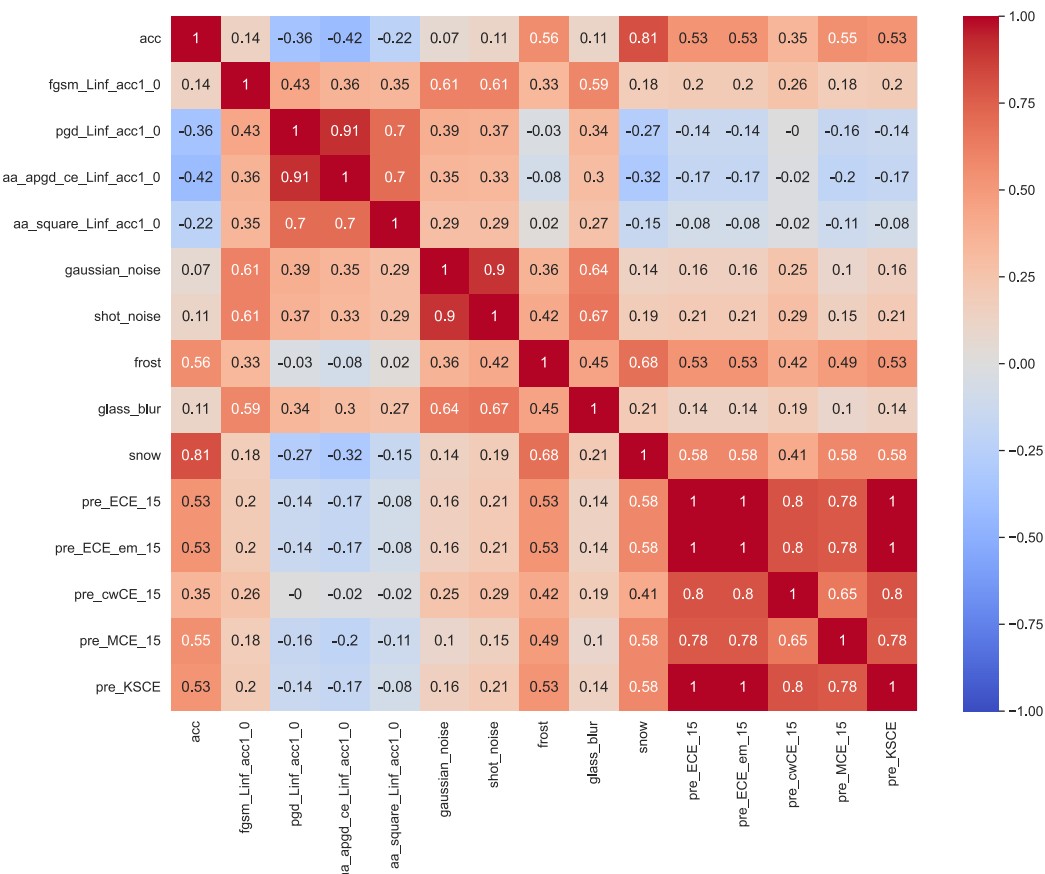

Figure 46: Kendall Ranking Correlation Matrix for the CIFAR-100 with calibration metrics measured on all models in TSS. The Correlation Matrix presents the ranking correlations between 5 different calibration metrics and various adversarial attack and image perturbation robustness measurements.

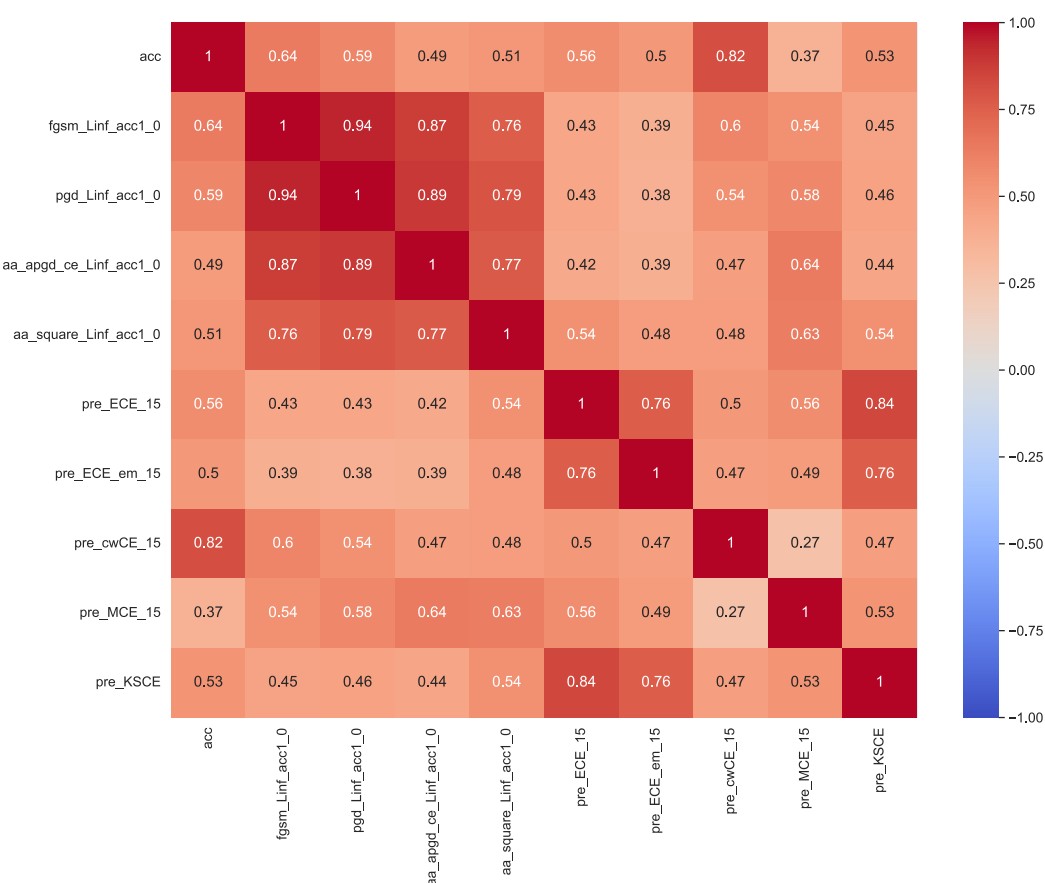

Figure 47: Kendall Ranking Correlation Matrix for the ImageNet16-120 with calibration metrics measured on TSS, filtered by top 1000 accuracy. The Correlation Matrix presents the ranking correlations between 5 different calibration metrics and various adversarial attack and image perturbation robustness measurements.

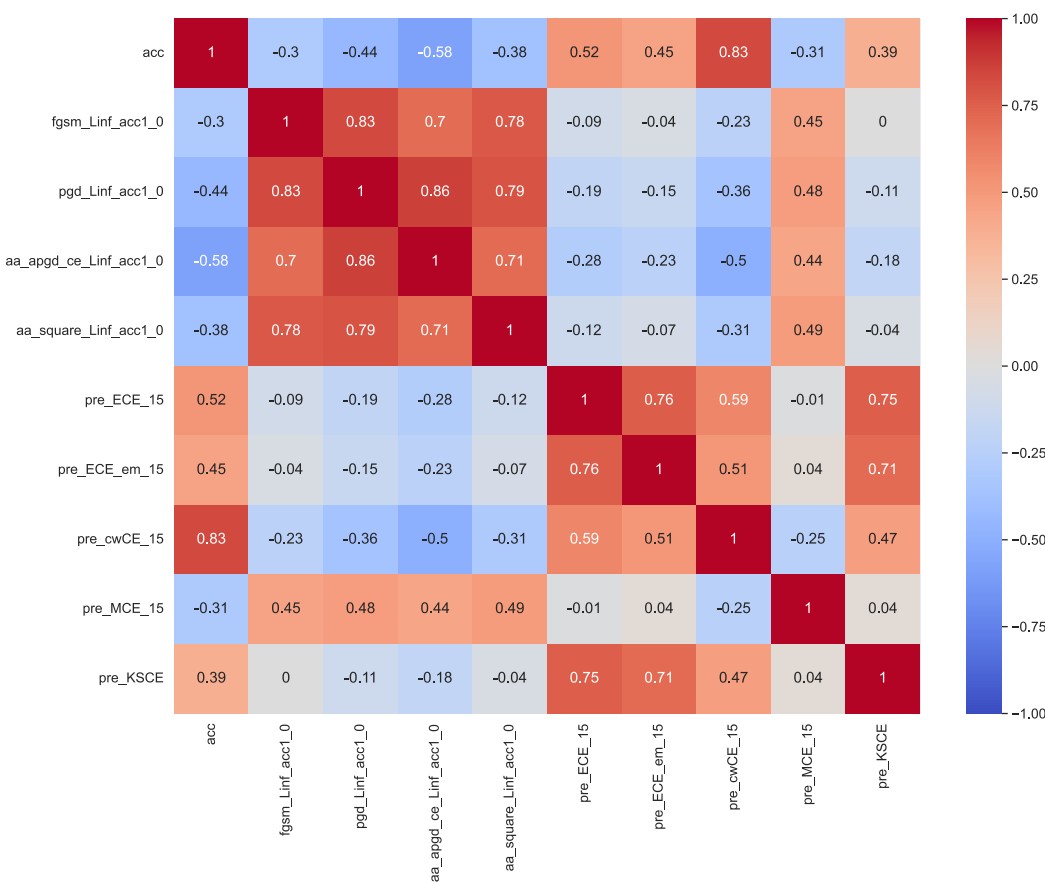

Figure 48: Kendall Ranking Correlation Matrix for the ImageNet16-120 with calibration metrics measured on TSS, filtered by top 5000 accuracy. The Correlation Matrix presents the ranking correlations between 5 different calibration metrics and various adversarial attack and image perturbation robustness measurements.

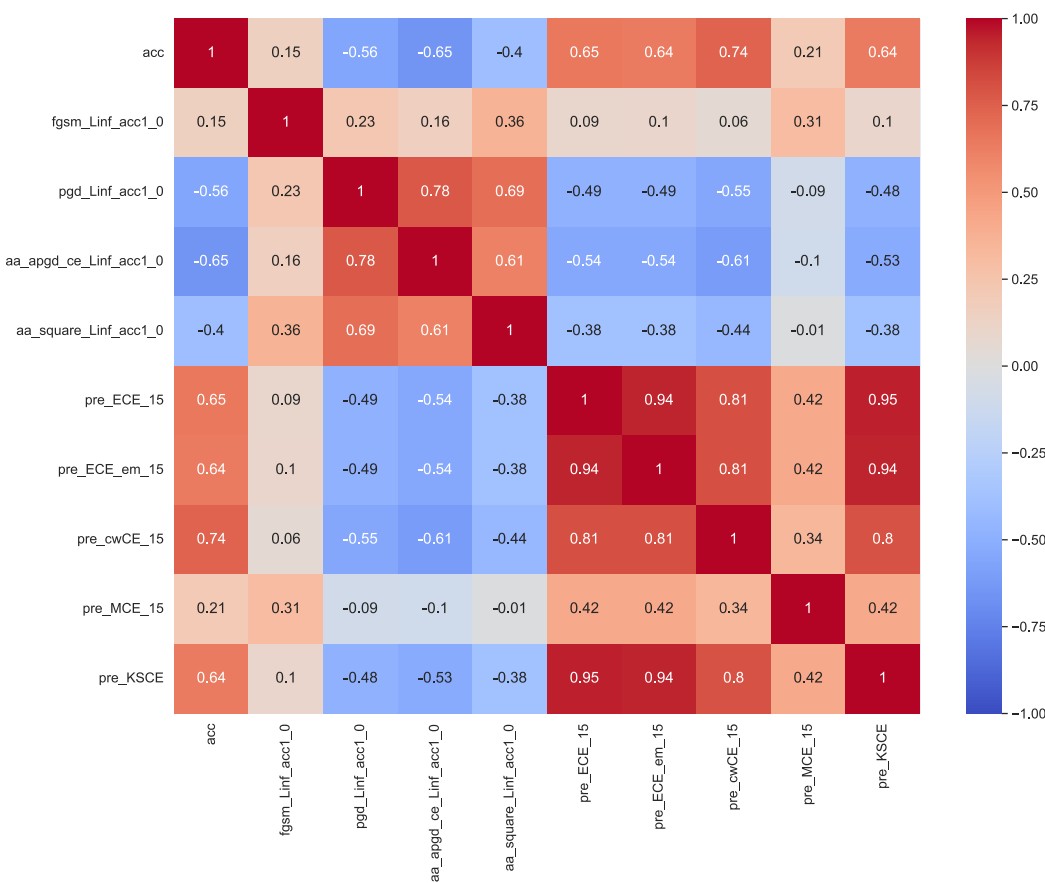

Figure 49: Kendall Ranking Correlation Matrix for the ImageNet16-120 with calibration metrics measured on all models in TSS. The Correlation Matrix presents the ranking correlations between 5 different calibration metrics and various adversarial attack and image perturbation robustness measurements.