# OpenReview forum: "A Benchmark Study on Calibration"
_ICLR.cc/2024/Conference — ICLR 2024 poster_

### Official Review · Reviewer_qRJE · 2023-10-31

**Soundness:** 3 good
**Presentation:** 3 good
**Contribution:** 3 good
**Rating:** 6
**Confidence:** 4

**Summary:**

The paper studies the calibration properties of deep neural networks (DNNs) using neural architecture search (NAS) search space. The motivation stems around the observation that, calibration properties of DNNs have not been thoroughly studied in the past, and NAS search space allows to create a comprehensive dataset of neural network architectures, which can be evaluated to study calibration properties. The dataset encompasses several bin-based and other calibration measurements across 117,702 unique neural network architectures. Particularly, the NATS-Bench has been used to curate the proposed dataset as it allows more broader search space, comprising models of various sizes. The study also includes eleven recent vision transformer architectures. The proposed analyses aims to answer seven different questions, including the interplay between accuracy and calibration, if calibration performance generalizes across datasets, the impact of bin sizes on calibration measurement, and which architectures are better for calibration. Post-hoc temperature scaling method is used as a calibration technique.

**Strengths:**

- The study of calibration properties of deep neural networks is an important research direction as it could allow developing well-calibrated architectures.

- The paper develops a comprehensive benchmark of neural network architectures that are then evaluated on different datasets to answer various questions. Further, recent vision transformer architectures have also been included as part of evaluation.

- Some questions included in the study are interesting and important: such as the Impact of bin sizes on calibration measurement and can model calibration be generalized across different datasets.

- Overall, the paper is well-written and it is not difficult to read and understand.

**Weaknesses:**

- Overall, the new questions posed and studied by the paper boils down to 1), 3) and 6) which are:

-- Model Calibration across different datasets

-- Reliability of calibration metrics

-- Impact of bin size on calibration metrics

- Other questions are mostly expansion of existing studies. This seems to undermine the overall contributions of the paper to some extent.

-Only post-hoc temperature scaling is used as a calibration technique to evaluate pre- and post calibration performance of a large chunk of models. There have been several new calibration methods, especially in train-time calibration paradigm, such as [A], [B] and [C].

-In 4.3: 1) What are the possible reasons of ECE showing consistent results with other metrics, although some other metrics are theoretically different? 2) Also, it is not clear that which calibration metric should be preferred over others in the scope of studies?

- What is the significance of 4th question (i.e. does a post-hoc calibration affect all models uniformly) in terms of advancing the research and algorithmic development in calibration?

- The abstract mentions ‘data pre-processing’ methods for improving calibration but such methods are discussed nowhere in the paper, including the related work.


[A] Liu, B., Ben Ayed, I., Galdran, A. and Dolz, J., 2022. The devil is in the margin: Margin-based label smoothing for network calibration. In Proceedings of the IEEE/CVF Conference on Computer Vision and Pattern Recognition (pp. 80-88).

[B] Patra, R., Hebbalaguppe, R., Dash, T., Shroff, G. and Vig, L., 2023. Calibrating deep neural networks using explicit regularisation and dynamic data pruning. In Proceedings of the IEEE/CVF Winter Conference on Applications of Computer Vision (pp. 1541-1549)

[C] Hebbalaguppe, R., Prakash, J., Madan, N. and Arora, C., 2022. A stitch in time saves nine: A train-time regularizing loss for improved neural network calibration. In Proceedings of the IEEE/CVF Conference on Computer Vision and Pattern Recognition (pp. 16081-16090).

**Questions:**

- How the proposed study and dataset would be helpful toward the development of new calibration methods for classification?

- It is a bit hard to grasp how the current analyses addresses the question of how reliable are calibration metrics?

- Would the finding that the bin size has a more substantial impact on Post-ECE be relevant for any train-time calibration technique too?

**Details Of Ethics Concerns:**

No ethical concerns could be identified.

---

> ### Author Response · Authors · 2023-11-13
>
> We appreciate the valuable comments from Reviewer qRJE
>
> **Re Expansion of existing studies:** We agree studies are the expansion of current research, however, they provide some different results from previous study. For example,
> - Previous works use AuC on OoD datasets  as a measurement of calibration[1][2]. However, in Section 4.2 we found that there is no correlation between the AuC and ECE, which indicate AuC on OoD may not be a reliable calibration measurement.
> - In section 4.4, we observe that a well calibrated model may not achieve better post-hoc calibration performance than a poor calibrated model. This observation supports the “calibratable” objective proposed in [1][2][3] in a way that research should focus more on obtaining a model with better post hoc calibration performance. Since the post-hoc calibration methods like temperature scaling are computationally cheap and effective, an overall calibration performance worth more attention. See more details in section "Re significance of Section 4.4".
> - Section 4.6 study the relationship between accuracy and calibration and section 4.7 study the architecture design for better calibration performance,  which are neither studied before from our understanding.
>
> **Re other calibration methods:** Following the suggestion, we incorporated Focal Loss[2] and MMCE Loss[11] as additional calibration methods. We trained six human-designed CNN models—ResNet18, ResNet34, ResNet50, ResNet110, Wide-ResNet-26-10, and Densenet121—on CIFAR-10 and CIFAR-100 using Focal Loss and MMCE Loss, both recognized as classic train-time calibration methods. Our observations, consistent with those reported in Appendix B, are detailed in Figure 16. As depicted in Figure 16 and similar to Figure 2, models trained using different train-time calibration methods all exhibit little correlation between results on CIFAR-10 and CIFAR-100. This suggests that the calibration properties of a specific architecture may not generalize effectively across different datasets. In Figure 17, ECE evaluated across different bin sizes shows little correlation between pre and post temperature scaling, indicating that well-calibrated models do not necessarily show improved calibration performance after post-hoc calibration techniques. This trend is particularly pronounced on CIFAR-100, where post-hoc calibration performance becomes negatively correlated with pre-calibration performance. Additionally, we observe that bin size can significantly impact post-hoc calibration performance, aligning with the observations in Section 4.4. In terms of the reliability of calibration metrics, we conducted an analysis of the correlation between all calibration metrics, as presented in Figure 14 in Appendix A and Figure 18 and Figure 19 in Appendix B. Notably, equal-mass classwise ECE exhibits a different pattern compared to other metrics, especially on CIFAR-100, reinforcing the observations outlined in Section 4.3.
>
>
> **Re consistency between metrics and preferred calibration metrics:**
> Although some other metrics are theoretically different from ECE, they share the same objective which aims to measure the alignment between the true likelihood and the predicted confidence. Thus, ECE shows consistent results with other metrics. However, this consistency is significantly influenced by the way how to approximate this alignment, such as the binning scheme. We mainly attribute the gap between class wise ECE and other metrics to the scenarios where the equal mass binning strategy aims to distribute samples uniformly across bins in a multi-class setting. This approach often leads to a scenario where the negative class is predominant. In practical applications of multiclass prediction, it's common for the model to assign very low confidence scores to a majority of the classes, often more than 95 out of 100. These low scores, hovering close to zero, indicate an absence of uncertainty. Equal mass class wise ECE will assign almost all bins to these near-zero confidence samples, except for one. Thus, equal mass binning experiences a significant decrease in estimation accuracy, making it unreliable. Thus, although all these metrics share the same objective, some of them cannot provide a precise estimation of this alignment in some scenarios. Considering this fact, we conduct the evaluation of the reliability of calibration metrics as in section 4.3, which we observe the unreliability of equal mass class wise ECE. On the other hand, the consistency between different calibration metrics with different theories such as ECE and MMCE support the reliability of metrics against each other. In the current stage, we recommend the use of calibration metrics that are more consistent with each other, such as ECE and MMCE.

---

> ### Author Response · Authors · 2023-11-13
>
> **Re significance of Section 4.4:** In Section 4.4, our observation highlights that a well-calibrated model with superior pre-calibration ECE may not necessarily achieve better post-hoc calibration performance than a poorly calibrated model with a higher pre-calibration ECE. For instance, in the table below, Model A exhibits a better pre-calibration ECE than Model B but offers less room for improvement through post-hoc methods, resulting in a poorer post-hoc ECE than Model B. Given that post-hoc calibration methods, such as temperature scaling, are both computationally efficient and effective, prioritizing an overall better calibration performance, or "calibratability," deserves more attention. Our observation supports the "calibratable" objective proposed in [1][2][3], suggesting that research efforts should prioritize achieving models with superior post-hoc calibration performance.
> | Model |  Accuracy%|Pre ECE%|Post ECE%|
> |--|--|--|--|
> | A | 95.05|2.994 |1.978|
> | B | 95.04| 3.893|1.107|
>
>
> **Re ‘data pre-processing’ methods:** By “data pre-processing”, we mean data augmentation methods such as mixup[6] and Augmix[7], we apologize for the confusion using “data pre-processing”.
>
> **Re help to develop to new calibration method:**
> - This paper can benefit the calibration research in different ways, for example:
> As an illustration, delving deeper into the architecture design reveals insights into achieving better calibration. For instance, as depicted in Figure 10, a discernible trend emerges, indicating that better-calibrated models exhibit a preference for Conv3*3 in edge 1 over Conv1*1 and favor a residual connection in edge 4 within the NATS-Bench scope. This implies that a better-calibrated model may lean towards incorporating larger kernel sizes in the early layers of a CNN block.
> - One can assess the validity of newly proposed metrics by evaluating their consistency with the set of metrics we have examined. Specifically, using the provided checkpoints and the 11 different metrics we assessed, a newly proposed metric can gauge its consistency by evaluating the checkpoints and calculating the Kendall ranking coefficient in comparison to the existing metrics. If the results exhibit little correlation or are negatively correlated with the established metrics, it suggests that the new metric warrants more careful analysis and scrutiny.
>
> **Re Analysis on reliability of calibration metrics:** We agree that current analyses are hard to measure the reliability of calibration metrics. However, this work reminds researchers to avoid using certain ambiguous metrics such as equal mass class-wise ECE  and AuC on OoD datasets.
>
> **Re bin-size impact on train-time calibration:** We conduct a toy experiments on bin-size involved train time calibration algorithm DECE[12] and observe that different bin size can bring larger impact on the post-hoc ECE, where small bin size works better for DECE, as shown in the table
> | Bin Size |  Accuracy%|Pre ECE%|Post ECE%|
> |--|--|--|--|
> | 2 | 95.85|3.994 |1.278|
> | 5 | 95.04| 3.893|1.107|
> | 10 | 95.07| 3.908|1.372|
> | 15 | 95.23 |  3.566| 1.361|
> | 50 | 94.96 |  4.031| 1.415|
> | 100 |95.02  |  3.943|1.523|
> | 200 |95.00  |  3.723|1.523|
> |  500|  94.97|  4.004|1.561|

---

> ### Author Response · Authors · 2023-11-13
>
> **Reference**
> ---
> [1] Sunil Thulasidasan, et al. On mixup training: Improved calibration and predictive uncertainty for deep neural networks. Advances in Neural Information Processing Systems, 32, 2019.
>
> [2]Mukhoti, Jishnu, et al. "Calibrating deep neural networks using focal loss." Advances in Neural Information Processing Systems 33 (2020): 15288-15299.
>
> [3] Wang, Deng-Bao, Lei Feng, and Min-Ling Zhang. "Rethinking calibration of deep neural networks: Do not be afraid of overconfidence." Advances in Neural Information Processing Systems 34 (2021): 11809-11820.
>
> [4] Wang, Deng-Bao, et al. "On the Pitfall of Mixup for Uncertainty Calibration." Proceedings of the IEEE/CVF Conference on Computer Vision and Pattern Recognition. 2023.
>
> [5] Calibration Bottleneck: What Makes Neural Networks less Calibratable? “ICLR 2024 Conference Submission3477”
>
> [6]Thulasidasan, Sunil, et al. "On mixup training: Improved calibration and predictive uncertainty for deep neural networks." Advances in Neural Information Processing Systems 32 (2019).
>
> [7] Hendrycks, Dan, et al. "Augmix: A simple data processing method to improve robustness and uncertainty." arXiv preprint arXiv:1912.02781 (2019).
>
>
> [8] Liu, B., Ben Ayed, I., Galdran, A. and Dolz, J., 2022. The devil is in the margin: Margin-based label smoothing for network calibration. In Proceedings of the IEEE/CVF Conference on Computer Vision and Pattern Recognition (pp. 80-88).
>
> [9] Patra, R., Hebbalaguppe, R., Dash, T., Shroff, G. and Vig, L., 2023. Calibrating deep neural networks using explicit regularisation and dynamic data pruning. In Proceedings of the IEEE/CVF Winter Conference on Applications of Computer Vision (pp. 1541-1549)
>
> [10] Hebbalaguppe, R., Prakash, J., Madan, N. and Arora, C., 2022. A stitch in time saves nine: A train-time regularizing loss for improved neural network calibration. In Proceedings of the IEEE/CVF Conference on Computer Vision and Pattern Recognition (pp. 16081-16090).
>
> [11] Kumar, Aviral, Sunita Sarawagi, and Ujjwal Jain. "Trainable calibration measures for neural networks from kernel mean embeddings." International Conference on Machine Learning. PMLR, 2018.
>
> [12] Bohdal O, Yang Y, Hospedales T. Meta-Calibration: Learning of Model Calibration Using Differentiable Expected Calibration Error[J]. arXiv preprint arXiv:2106.09613, 2021.

---

> ### Author Response · Authors · 2023-11-21
>
> Respected reviewer, should you have any further concerns, I am eagerly anticipating your response.

---

> ### Comment · Reviewer_qRJE · 2023-11-22
>
> I thanks authors for responding to my comments. The responses to a majority of comments are satisfactory, including evaluation with train-time calibration methods, bin size impact, and significance of sec. 4.4. However, the following points require better explanation: 1) how the proposed study can encourage new research in this regard, and on the 2) expansion of existing methods. For instance, 1) is mostly missing description about the potential of developing new calibration methods with this study.

---

> > ### Author Response · Authors · 2023-11-22
> >
> > Dear qRJE,
> >
> > Thank you for your thoughtful feedback on our manuscript. We appreciate the positive remarks regarding our responses to the majority of your comments and are grateful for the opportunity to address the concerns you raised.
> >
> > ## How the proposed study can encourage new research in this regard?
> > 1. Most of current researches such as [1][2][3] focus on calibrating the model on the fixed datasets. From Sec 4.1, we observe the calibration property of a certain architecture can not generalize well to different datasets, which means the model can well-calibrated on CIFAR10 but poor calibrated on ImageNet. The generalizability of calibration is not well studied in the literature. We believe this is an important problem to be solved. For example, we can consider the calibration as a ensemble problem, and try to find a way to ensemble the calibration property of different datasets.
> >
> > 2. The choice of bin size for calibration error measurement introduces a bias-variance tradeoff, influencing the calibration estimate's quality. As elucidated in Section 4.6, our findings indicate that post-temperature scaling calibration error is sensitive to the number of bins, raising questions about the reliability of different bin sizes in assessing a model's calibration. This prompts a crucial research question: how can we systematically select the number of bins for post-temperature-scaling ECE calibration error measurement to obtain a more reliable evaluation of a model's calibration property?
> >
> > 3. From Sec 4.2, we observe that robustness measurement may not align well with calibration performance, which means a model with good robustness performance may have high calibration error. As robustness and calibration are both important factors towards trustworthty machine learning. The tradeoff between robustness and calibration is not well studied in the literature. We believe this is an important problem to be solved. For example, we can consider the calibration and robustness as a multi-objective optimization problem, and try to apply multi-objective algorithms, such as NSGA[4] to balance the tradeoff between robustness and calibration and study the Pareto frontier of the tradeoff.
> >
> >
> >
> > ## How the proposed study can encourage the expansion of existing methods?
> > 1. Traditional ensemble methods [5][6][7] often aim to enhance calibration on fixed datasets. Building upon our observation in Section 4.1, where the calibration property of a given architecture lacks generalizability across datasets, we suggest a novel approach: combining calibrated ensemble members from different datasets to enhance the overall calibration generalizability of the model.
> >
> > 2. Current train-time calibration methods like SoftECE [8] and DECE [9] use 15-bin pre-temperature scaling ECE as an auxiliary calibration objective. However, as shown in Figure 4b, post-temperature scaling ECE exhibits little correlation with pre-temperature scaling ECE, and different bin sizes yield distinct calibration performance measurements. This prompts us to propose a potential expansion of existing methods by considering both post-temperature scaling ECE and ECE of varying bin sizes as auxiliary calibration objectives.
> >
> >
> >
> > ---
> > [1] Mukhoti, Jishnu, et al. "Calibrating deep neural networks using focal loss." Advances in Neural Information Processing Systems 33 (2020): 15288-15299.
> >
> > [2] Tao, Linwei, Minjing Dong, and Chang Xu. "Dual Focal Loss for Calibration." arXiv preprint arXiv:2305.13665 (2023).
> >
> > [3] Liu, Bingyuan, et al. "The devil is in the margin: Margin-based label smoothing for network calibration." Proceedings of the IEEE/CVF Conference on Computer Vision and Pattern Recognition. 2022.
> >
> > [4] Deb, Kalyanmoy, et al. "A fast and elitist multiobjective genetic algorithm: NSGA-II." IEEE transactions on evolutionary computation 6.2 (2002): 182-197.
> >
> > [5] Zhang, Jize, Bhavya Kailkhura, and T. Yong-Jin Han. "Mix-n-match: Ensemble and compositional methods for uncertainty calibration in deep learning." International conference on machine learning. PMLR, 2020.
> >
> > [6] Zou, Yuli, Weijian Deng, and Liang Zheng. "Adaptive Calibrator Ensemble for Model Calibration under Distribution Shift." arXiv preprint arXiv:2303.05331 (2023).
> >
> > [7] Xiong, Ruibin, et al. "Uncertainty calibration for ensemble-based debiasing methods." Advances in Neural Information Processing Systems 34 (2021): 13657-13669.
> >
> > [8] Karandikar, Archit, et al. "Soft calibration objectives for neural networks." Advances in Neural Information Processing Systems 34 (2021): 29768-29779.
> >
> > [9] Wang, Cheng, and Jacek Golebiowski. "Meta-Calibration Regularized Neural Networks." arXiv preprint arXiv:2303.15057 (2023).

---

> ### Author Response · Authors · 2023-11-23
>
> Respected reviewer, should you have any further concerns, I am eagerly anticipating your response.

---

### Official Review · Reviewer_Xzoe · 2023-11-01

**Soundness:** 3 good
**Presentation:** 3 good
**Contribution:** 2 fair
**Rating:** 3
**Confidence:** 4

**Summary:**

The paper presents a study that analyzes the relationship between NAS and calibration powers of neural networks. The paper combines CIFAR-10, CIFAR-100, and ImageNet as the dataset in which multiple architectures are tested and measure its calibration powers. The study of the paper thus focuses only on image classification problems using small- and medium-scale datasets.

**Strengths:**

S1. I really value the topic of calibration as I believe it is a good feature to have in many classification tasks and systems. I think the paper tackles an important problem.

S2. The clarity of the paper is good, the narrative flows well, and is easy to understand and follow.

**Weaknesses:**

W1. The motivation about why NAS + Calibration is important is missing in the paper. Unfortunately, the paper lacks a clear justification for studying NAS + Calibration. It is not clear intuitively why this is a good direction to explore. It is not clear why a wholistic approach is not worth exploring over NAS + Calibration. Unfortunately, the paper makes the reader believe that the analysis was done just because it has not done before. I think the paper really needs to justify why NAS + Calibration is a good angle to study.

W2. The study uses small- and medium- scale datasets for analyzing the relationship between NAS + Calibration. In particular, the study uses CIFAR-10, CIFAR-100, and ImageNet datasets for the analysis. While I understand that small datasets are easier to handle given the NAS component of the study, it is questionable the conclusions one can get from these small datasets. While ImageNet is larger, compared to modern large-scale datasets, such as, LAION, its use is also questionable. Modern image classification methods are trained using foundation models that use really large-scale datasets (e.g., LAION) and those are the worth studying in my opinion since they are the ones adopted in industry and are making an impact. I think the paper needs to justify the use of these small- and medium-scale datasets. Otherwise, the conclusions drawn from the studies are not solid.

**Questions:**

Please see the Weaknesses stated above.

---

> ### Author Response · Authors · 2023-11-13
>
> We appreciate the valuable comments from Reviewer Xzoe
>
>
> **Re NAS + Calibration:** To avoid potential misunderstanding regarding the term "NAS + Calibration", please let us clarify that this study does not involve conducting neural architecture search (NAS) specifically for calibration purposes. The primary focus of this research is the examination of calibration properties. To address calibration-related research questions, such as the reliability of calibration metrics, one approach is to assess the consistency of different metrics based on a substantial number of well-trained models. However, collecting such a substantial dataset is often challenging due to the associated training costs. Fortunately, NATS-Bench [1] provides access to 117.9K well-trained models with various architectural designs, enabling us to conduct a comprehensive and generalisable study.
>
> **Re datasets:**
> Modern large-scale datasets, such as LAION, primarily serve image-text alignment tasks, notably in training stable diffusion models. However, it's worth noting that calibration tasks typically revolve around classification and regression. In recent years, standard benchmarks for calibration studies have included well-known datasets like CIFAR-10, CIFAR-100, and ImageNet [2] [3] [4] [5]. While the prospect of applying calibration tasks to LAION is intriguing, it falls outside the scope of the present work.
>
> ---
> [1] Dong, Xuanyi, et al. "Nats-bench: Benchmarking nas algorithms for architecture topology and size." IEEE transactions on pattern analysis and machine intelligence 44.7 (2021): 3634-3646.
>
> [2] Tao, Linwei, Minjing Dong, and Chang Xu. "Dual Focal Loss for Calibration." arXiv preprint arXiv:2305.13665 (2023).
>
> [3] Ghosh, Arindam, Thomas Schaaf, and Matthew Gormley. "AdaFocal: Calibration-aware Adaptive Focal Loss." Advances in Neural Information Processing Systems 35 (2022): 1583-1595.
>
> [4] Minderer, Matthias, et al. "Revisiting the calibration of modern neural networks." Advances in Neural Information Processing Systems 34 (2021): 15682-15694.
>
> [5] Gawlikowski, Jakob, et al. "A survey of uncertainty in deep neural networks." Artificial Intelligence Review 56.Suppl 1 (2023): 1513-1589.

---

> > ### Comment · Reviewer_Xzoe · 2023-11-21
> >
> > **NAS + Calibration**: "The primary focus of this research is the examination of calibration properties" This is not clear in the narrative. Please fix.
> >
> > **Datasets**:
> > - "Modern large-scale datasets, such as LAION, primarily serve image-text alignment tasks, notably in training stable diffusion models" This is not true. LAION has also been used to train CLIP-like models, which end up being the foundation of image classifiers (including zero-shot classification models); see https://github.com/mlfoundations/open_clip.
> >
> > - " [...] standard benchmarks for calibration studies have included well-known datasets like CIFAR-10, CIFAR-100, and ImageNet [...]" Unfortunately, these datasets are so tiny compared to the ones one can use to evaluate image classification in modern days. I would not trust any experiment in modern days using these datasets because, as shown recently, data is very important (see CLIP, DALLE, etc.). Moreover, in many ways, the more data the better the estimates and the conclusions. Statistically speaking, since many image classifiers optimize a loss based on statistical foundations, the size of the dataset matters. Thus, I don't think this fall outside the scope of the paper. I think modern papers should work with large-scale datasets since many working in real-world scenarios were trained w/ large datasets, not tiny ones.

---

> ### Author Response · Authors · 2023-11-21
>
> Respected reviewer, should you have any further concerns, I am eagerly anticipating your response.

---

> ### Author Response · Authors · 2023-11-22
>
> Dear Reviewer Xzoe,
>
> Thank you for taking the time to review our paper. We appreciate your valuable feedback and suggestions. Below are our responses to the points you raised:
>
> ## 1. Calibration Focus:
> **Original Comment:** "The primary focus of this research is the examination of calibration properties. This is not clear in the narrative. Please fix."
>
> **Revised Response:** We have address this concern by refining our statement to explicitly state that the primary objective of our research is to investigate calibration properties, as modified in the fourth paragraph of introduction.
>
>
> ## 2. Datasets Section:
>
> In our examination of 249 recent calibration papers, we discovered that none of them—0 out of 249—involved training or evaluating models on LAION. The fact that most recent calibration papers are not using LAION, which indicates that LAION is not a widely used dataset in the calibration community, and using widely adopted CIFAR10, CIFAR100 and ImageNet can prove the effectiveness of calibration algorithms as other accepted calibration papers[2][3][4][5].
>
> Specifically, to delve into the use of large-scale datasets, specifically LAION, in calibration research, we scrutinized 249 calibration-related papers across ICLR2023 submissions, ICLR2024 submissions, NIPS2023 accepted papers, and ICML2023 accepted papers. The criterion for calibration-related papers was the inclusion of the keywords "calibration," "confidence," or "uncertainty" in the title. Our investigation revealed that one[1] of these papers cited LAION, and none utilized LAION for experiments. The web crawler code can be found in this [anonymous code link](https://anonymous.4open.science/r/ICLR2024-rebuttal-F7D4/). The detailed statistics are shown in the table below.
>
> | | ICLR2023 | ICLR2024 | NIPS2023 | ICML2023 |
> |:---:|:---:|:---:|:---:|:---:|
> | # of total papers | 4920 | 7252 | 3217 | 1828 |
> | # of calibration-related papers | 63 | 107 | 49 | 30 |
> | # of calibration-related papers that involved LAION | 0 | 1 | 0 | 0 |
> | Paper |/ | [1] | / | / |
>
>
> If you have further concern, please don't hesitate to comment.
>
> Thank you once again for your insightful review.
>
> Best regards,
>
> Submission 1053 Authors
>
> ## References
> [1] CONFIDENCE-AWARE REWARD OPTIMIZATION FOR FINE-TUNING TEXT-TO-IMAGE MODELS (ICLR2024 submission)
>
> [2] Tao, Linwei, Minjing Dong, and Chang Xu. "Dual Focal Loss for Calibration." arXiv preprint arXiv:2305.13665 (2023).
>
> [3] Ghosh, Arindam, Thomas Schaaf, and Matthew Gormley. "AdaFocal: Calibration-aware Adaptive Focal Loss." Advances in Neural Information Processing Systems 35 (2022): 1583-1595.
>
> [4] Minderer, Matthias, et al. "Revisiting the calibration of modern neural networks." Advances in Neural Information Processing Systems 34 (2021): 15682-15694.
>
> [5] Gawlikowski, Jakob, et al. "A survey of uncertainty in deep neural networks." Artificial Intelligence Review 56.Suppl 1 (2023): 1513-1589.

---

> ### Comment · Reviewer_Xzoe · 2023-11-22
>
> **Datasets Section**: While previous papers use these tiny datasets, I as a reviewer, object to use them in modern research. The reason is that statistically speaking, they are small, and as many other systems nowadays have shown (e.g., CLIP, OpenCLIP, etc..), large and high-quality datasets are important. To me, the use of these datasets do not represent the real scenarios of machine learning; mainly because their image resolution is quite small, their scale is tiny, and the scenarios they cover are niche. If we don't aim to show results on large-scale datasets, I don't expect the field in making leaps forward in understanding machine learning.

---

> > ### Author Response · Authors · 2023-11-22
> >
> > Dear Reviewer Xzoe,
> >
> > In response to your comments, we would like to emphasize the following points:
> >
> > 1. Admittedly, using larger datasets that approximate real-world scenarios can yield results that are closer to reality. LAION, being a substantial dataset, raises questions about the resources involved, such as the number of GPUs and hours required for training (For instance, ViT-B/16 was trained with 176 A100 (40 GB) GPUs for approximately 61 hours, totaling 10,700 GPU-hours, while ViT-L/14 utilized 400 A100 (40 GB) GPUs for around 127 hours, totaling 50,800 GPU-hours [1]). Up to this point, the majority of work on LAION has been conducted by industry players [2][3][4] and remains financially out of reach for most university research groups.
> >
> >     In our perspective, the roles of industry and academia differ in the modern research era. In the age of large models, industry focuses on scaling up algorithms and deploying them in real-world applications. On the other hand, universities engage in exploratory research, often staying at the level of effective prototypes and forward-thinking ideas. This distinction is significant, as industry work must demonstrate practical value, while university research, with its focus on novel concepts, remains valuable in its own right. A notable example is the original diffusion model[5], initially proposed by academia. If stable diffusion [4], trained on LAION, had not acknowledged the modest initial results of the original diffusion model [5] on a smaller dataset (CIFAR-10), the landscape of present-day generative models might not have taken shape, thereby highlighting the intrinsic value of university's pioneering research.
> >
> > 2. The absence of LAION's dataset in other papers does not necessarily diminish their contributions or render them unworthy of acceptance (e.g., [6][7][8][9]). Even in the absence of LAION, these papers, through fair comparisons with alternative algorithms on relatively smaller datasets, demonstrate the effectiveness, completeness, and insightful contributions of their proposed approaches. Therefore, the presence of LAION should not be the sole determinant of a paper's worth or contribution.
> > ---
> >
> > ### References
> >
> > [1] https://github.com/mlfoundations/open_clip/blob/main/docs/PRETRAINED.md
> >
> > [2] Sauer, Axel, et al. "Stylegan-t: Unlocking the power of gans for fast large-scale text-to-image synthesis." arXiv preprint arXiv:2301.09515 (2023).
> >
> > [3] Li, Yanghao, et al. "Scaling language-image pre-training via masking." Proceedings of the IEEE/CVF Conference on Computer Vision and Pattern Recognition. 2023.
> >
> > [4] Rombach, Robin, et al. "High-resolution image synthesis with latent diffusion models." Proceedings of the IEEE/CVF conference on computer vision and pattern recognition. 2022.
> >
> > [5] Jascha Sohl-Dickstein, Eric A. Weiss, Niru Mah- eswaranathan, and Surya Ganguli. Deep unsupervised learning using nonequilibrium thermodynamics. CoRR, abs/1503.03585, 2015. 1, 3, 4, 18
> >
> > [6] Tao, Linwei, Minjing Dong, and Chang Xu. "Dual Focal Loss for Calibration." arXiv preprint arXiv:2305.13665 (2023).
> >
> > [7] Ghosh, Arindam, Thomas Schaaf, and Matthew Gormley. "AdaFocal: Calibration-aware Adaptive Focal Loss." Advances in Neural Information Processing Systems 35 (2022): 1583-1595.
> >
> > [8] Minderer, Matthias, et al. "Revisiting the calibration of modern neural networks." Advances in Neural Information Processing Systems 34 (2021): 15682-15694.
> >
> > [9] Gawlikowski, Jakob, et al. "A survey of uncertainty in deep neural networks." Artificial Intelligence Review 56.Suppl 1 (2023): 1513-1589.

---

> > > ### Comment · Reviewer_Xzoe · 2023-11-22
> > >
> > > 1. I understand academia is facing challenges due to the lack of resources.
> > > 2. My overall concern in sum is that it is unclear how we can ensure that the conclusions derived from the tiny and niche datasets generalizes well to other domains. This is why I recommended a larger dataset (it is not the only one, for example, Google Open Images is another good dataset) since that is guaranteed to cover several aspects of visual recognition. In short, I am concerned that the impact of this paper is minimal due to using tiny and niche datasets.

---

> > > > ### Author Response · Authors · 2023-11-23
> > > >
> > > > Dear Reviewer Xzoe,
> > > >
> > > > We would like to inform you that we have conducted preliminary experiments to explore the calibration properties of models pretrained on large modern datasets, including LAION-5b, LAION-400m, and YFCC100M. Our initial findings closely align with the results outlined in the main text, with additional details provided in Appendix C.
> > > >
> > > > In these experiments, we assessed the performance of seven LAION-5b pretrained models on ImageNet-1K. Furthermore, we investigated the zero-shot calibration capabilities of seven CLIP models pretrained on larger datasets, evaluating their performance on CIFAR10 and CIFAR100, respectively.
> > > >
> > > > **Key insights from our experiments include:**
> > > >
> > > > 1. The limited generalizability of calibration properties across different datasets, as discussed in section 4.1.
> > > >
> > > > 2. The unreliability of equal-mass classwise ECE as a metric, as emphasized in section 4.3.
> > > >
> > > > 3. The substantial impact of bin size on post-temperature ECE, elaborated on in section 4.4.
> > > >
> > > > 4. Well-calibrated models do not necessarily exhibit better calibration performance after post-hoc calibration techniques as discussed in section 4.4.
> > > >
> > > > We value your attention to our work and eagerly anticipate any feedback or suggestions you may have.
> > > >
> > > > **Best regards,**
> > > >
> > > > Submission 1053 Authors

---

> > > > > ### Comment · Reviewer_Xzoe · 2023-11-30
> > > > >
> > > > > Thanks for doing these experiments. However, I do question the use of pre-trained models using large dataset on small datasets. In other words, my concern is that the testing bed (CIFAR 10, CIFAR100, etc.) is small, not challenging, and not that realistic. Perhaps a more meaningful experiment is using ImageNet as the testing set, and re-measure the performance.

---

### Official Review · Reviewer_S1Jc · 2023-11-03

**Soundness:** 2 fair
**Presentation:** 2 fair
**Contribution:** 2 fair
**Rating:** 6
**Confidence:** 3

**Summary:**

This paper conducts several investigations about the calibration problem of the deep neural networks. This paper introduces a calibration dataset based on the NATS-Bench for generating massive CNNs with different topologies or model sizes. This paper adopts the calibration dataset and evaluate the different calibration metrics to analyze the calibration properties in deep neural networks. This paper provides extensive explorations and conclusions through the benchmarks

**Strengths:**

1.	This paper raises 7 initial questions for exploring the calibration properties in deep neural networks.
2.	This paper builds a benchmark based on models generated by NAS for evaluating the calibration metrics.
3.	This paper conducts extensive experiments to analyze and answer the questions.

**Weaknesses:**

1.	The proposed calibration benchmark might be limited since it contains only convolution neural networks for image classification. I’m concerned about how about the calibration properties for other tasks, e.g., object detection or NLP tasks. It’s more convincing when extending the benchmark for more architectures and more tasks.
2.	Most architectures and networks are generated from the similar search space, which might have similar effects and are limited for the conclusions. Varying the search spaces and using human designed architectures are necessary.
3.	Different architectures on different benchmarks/dataset might require different hyper-parameters. Though it’s hard to search optimal/sub-optimal parameters for different models, it will affect the experimental results.
4.	This paper explores the calibration properties in neural networks, but I’m much concerned about how those findings impact the future research or guide the designing process for both accuracy and calibration performance.

**Questions:**

See the weaknesses above

---

> ### Author Response · Authors · 2023-11-13
>
> We appreciate the valuable comments from Reviewer S1Jc
>
> **Re findings generality:** While our analysis in the main text primarily focuses on CNN models, we extend our investigation to transformer-based models, as detailed in Appendix A. In further support of our study, we trained six human-designed CNN models—ResNet18, ResNet34, ResNet50, ResNet110, Wide-ResNet-26-10, and Densenet121—on CIFAR-10 and CIFAR-100, obtaining similar conclusions as reported in Appendix B. As depicted in Figure 12 in Appendix A and Figure 17 in Appendix B, the ECE across different bin sizes demonstrates little correlation between pre and post temperature scaling for both Transformers and CNNs. This suggests that well-calibrated models do not necessarily exhibit improved calibration performance after post-hoc calibration techniques. This phenomenon is particularly pronounced on CIFAR-100, where post-hoc calibration performance becomes negatively correlated with pre-calibration performance. Regarding the impact of bin size, we observe a substantial influence on post-hoc calibration performance, aligning with the observations outlined in Section 4.4. In terms of the reliability of calibration metrics, we conducted an analysis of the correlation between all calibration metrics, as presented in Figure 14 in Appendix A and Figure 18 and Figure 19 in Appendix B. Notably, equal-mass classwise ECE exhibits a different pattern compared to other metrics, especially on CIFAR-100, reinforcing the observations outlined in Section 4.3.
>
> **Re NLP Task:** Recognizing the significance of task diversification, we undertook an NLP classification experiment using the 20 newsgroups dataset. Our experimental setup included 2 CNN-based models, 1 MLP-based model, 2 RNN models, and 1 BERT model. Each model is trained for 20 epochs. Notably, the correlation among the six models between pre-calibration ECE and post-hoc ECE was found to be -0.13. This aligns with the observation in Figure 4b, indicating that the impact of post-hoc calibration methods is not uniform across all models.
>
> **Re NAS search space:** In an effort to extend the applicability of our observations beyond a limited search space, we conducted experiments on human-designed Transformers and CNNs. The outcomes of these experiments align with the findings in the main text, as detailed in Appendices A and B.
>
>
> **Re hyper-parameters fine tune:** While we acknowledge that hyperparameter tuning for each model could potentially yield more precise results, the vast number of models involved in our study (117,702 unique neural networks) makes this approach computationally intensive. As a pragmatic tradeoff, we opted to establish a fixed and fair set of hyperparameters for all models to conduct our experiments.
>
> **Re border impact:** Our several observations can help improve the research in calibration and accuracy, for example,
> - In section 4.1, we suggest testing model calibration performance on downstream tasks specifically, since the fact that a well calibrated model on source tasks does not necessarily perform well on downstream tasks.
> - In section 4.2 and 4.3, we suggest avoiding equal mass class wise ECE or AuC on OoD datasets for calibration evaluation, which helps the calibration evaluation.
> - In section 4.4, we observe that a well calibrated model may not achieve better post-hoc calibration performance than a poor calibrated model. This observation supports the “calibratable” objective proposed in [1][2][3] in a way that research should focus more on obtaining a model with better post hoc calibration performance.
> - In section 4.5, we observe the tradeoff between accuracy and ECE, which can facilitate the research in prediction accuracy.
> - In section 4.6, the impact of bin-size might support the study of those train-time calibration methods which involve the hyperparameter bin size, such as SoftECE[4] and DECE[5].
> - In section 4.7, the analysis on architecture design of better calibration models can support the research on calibration from the view of architecture design.
>
>
> ---
> [1] Wang, Deng-Bao, Lei Feng, and Min-Ling Zhang. "Rethinking calibration of deep neural networks: Do not be afraid of overconfidence." Advances in Neural Information Processing Systems 34 (2021): 11809-11820.
>
> [2] Wang, Deng-Bao, et al. "On the Pitfall of Mixup for Uncertainty Calibration." Proceedings of the IEEE/CVF Conference on Computer Vision and Pattern Recognition. 2023.
>
> [3] Calibration Bottleneck: What Makes Neural Networks less Calibratable? “ICLR 2024 Conference Submission3477”
>
> [4] Karandikar A, Cain N, Tran D, et al. Soft calibration objectives for neural networks[J]. Advances in Neural Information Processing Systems, 2021, 34: 29768-29779.
>
> [5] Bohdal O, Yang Y, Hospedales T. Meta-Calibration: Learning of Model Calibration Using Differentiable Expected Calibration Error[J]. arXiv preprint arXiv:2106.09613, 2021.

---

> ### Author Response · Authors · 2023-11-21
>
> Respected reviewer, should you have any further concerns, I am eagerly anticipating your response.

---

### Meta-Review · Area_Chair_EtDH · 2023-12-05

**Metareview:**

Model calibration is an important problem, and authors have investigated different calibration properties. One limitation is that the results are only on image classification. However, overall the contribution from the draft is positive.

The concern of one reviewer regarding results not being or large dataset is valid, however one can understand that the computational cost associated with such experiments could be restricting. Where results on large datasets are becoming more common, we have to empathize with how this will restrict the research, especially in academia. We will encourage the authors to include discussions, results, and analysis contributed during the rebuttal, in the final draft. Also please add a few lines regarding why contributions are valid even without results on a much larger dataset.

regards
meta reviewer

**Justification For Why Not Higher Score:**

Positive reviews are at max 6.

**Justification For Why Not Lower Score:**

Positive reviews are at max 6.

---

### Decision · Program_Chairs · 2024-01-16

Accept (poster)